# Effector loss drives adaptation of *Pseudomonas syringae* pv. *actinidiae* biovar 3 to *Actinidia arguta*

Lauren M. Hemara[1,2,3‡], Jay Jayaraman[1,3‡], Paul W. Sutherland[1], Mirco Montefiori[1], Saadiah Arshed[1], Abhishek Chatterjee[1], Ronan Chen[4], Mark T. Andersen[1], Carl H. Mesarich[3,5], Otto van der Linden[1], Minsoo Yoon[1], Magan M. Schipper[6], Joel L. Vanneste[6], Cyril Brendolise[1], Matthew D. Templeton[1,2,3]*

1 The New Zealand Institute for Plant and Food Research Limited, Mt. Albert Research Centre, Auckland, New Zealand, 2 School of Biological Sciences, University of Auckland, Auckland, New Zealand, 3 Bioprotection Aotearoa, New Zealand, 4 The New Zealand Institute for Plant and Food Research Limited, Food Industry Science Centre, Palmerston North, New Zealand, 5 School of Agriculture and Environment, Massey University, Palmerston North, New Zealand, 6 The New Zealand Institute for Plant and Food Research Limited, Ruakura Campus, Hamilton, New Zealand

‡ These authors share first authorship on this work.
* matt.templeton@plantandfood.co.nz, m.templeton@auckland.ac.nz

**Data Availability Statement:** All relevant data are within the manuscript and its Supporting Information files.

## Abstract

A pandemic isolate of *Pseudomonas syringae* pv. *actinidiae* biovar 3 (Psa3) has devastated kiwifruit orchards growing cultivars of *Actinidia chinensis*. In contrast, *A. arguta* (kiwiberry) is not a host of Psa3. Resistance is mediated via effector-triggered immunity, as demonstrated by induction of the hypersensitive response in infected *A. arguta* leaves, observed by microscopy and quantified by ion-leakage assays. Isolates of Psa3 that cause disease in *A. arguta* have been isolated and analyzed, revealing a 51 kb deletion in the exchangeable effector locus (EEL). This natural EEL-mutant isolate and strains with synthetic knockouts of the EEL were more virulent in *A. arguta* plantlets than wild-type Psa3. Screening of a complete library of Psa3 effector knockout strains identified increased growth *in planta* for knockouts of four effectors–AvrRpm1a, HopF1c, HopZ5a, and the EEL effector HopAW1a – suggesting a resistance response in *A. arguta*. Hypersensitive response (HR) assays indicate that three of these effectors trigger a host species-specific HR. A Psa3 strain with all four effectors knocked out escaped host recognition, but a cumulative increase in bacterial pathogenicity and virulence was not observed. These avirulence effectors can be used in turn to identify the first cognate resistance genes in *Actinidia* for breeding durable resistance into future kiwifruit cultivars.

## Author summary

Clonally propagated monoculture crop plants facilitate the emergence and spread of new diseases. Plant pathogens cause disease by the secretion of effectors that function by repressing the host defense response. While the last few decades have seen a huge increase

**Funding:** This work was funded by Bio-protection Aoteoroa (Tertiary Education Commission, New Zealand) to MDT, including a post-doctoral fellowship to JJ, and a Rutherford Foundation Post-doctoral fellowship (Royal Society of New Zealand) to JJ. LMH would like to thank Zespri International for an MSc scholarship. The funders had no role in study design, data collection and analysis, decision to publish, or preparation of the manuscript.

**Competing interests:** The authors have declared that no competing interests exist.

in our understanding of the role effectors play in mediating plant-pathogen interactions, the combinations of effectors required for the establishment of plant disease and that account for host specificity are less well understood. Breeding genetic resistance is often used to protect plants from disease but it is frequently evaded by rapidly evolving pathogens. *Pseudomonas syringae* pv. *actinidiae* (Psa) which causes bacterial canker disease of kiwifruit has spread rapidly throughout the world's kiwifruit orchards, particularly those growing cultivars of *Actinidia chinensis*. Other Actinidia species including *A. arguta* display strong resistance conferred by recognition of effectors delivered by Psa. We explore the depth and dynamics of Psa effector recognition by *A. arguta* and show that there is a trade-off between losses of effector recognition by *A. arguta* versus the retention of pathogenicity. Our findings should aid in the understanding of how to breed durable resistance into perennial plants challenged by swiftly evolving pathogens.

## Introduction

The *Pseudomonas syringae* species complex contains over 60 pathovars, each with a discrete host range [1–3]. The collective host breadth of the *P. syringae* species complex makes this bacterial plant pathogen an ideal model for studying the molecular basis of host specificity. *P. syringae* pv. *actinidiae* (Psa), the causal agent of kiwifruit canker, is a recently emerged plant pathogen. The disease was first isolated from *A. chinensis* var. *deliciosa* (green-fleshed kiwifruit), and *A. arguta* (kiwiberry) in Japan in 1984 [4,5]. There was a subsequent outbreak in South Korea in the mid-1990s [6]. However, it was the emergence of a pandemic strain that spread rapidly around the world from 2008, which particularly devastated orchards of *Actinidia chinensis* var. *chinensis* (gold-fleshed kiwifruit) [7,8]. Isolates from these three separate outbreaks of bacterial canker have been grouped into biovars and recently two more biovars have been described [9,10]. Biovars of Psa have closely related core genes and are primarily distinguished by their variable accessory genomes, which include effectors and toxin biosynthesis clusters [11].

Psa was first detected in New Zealand's kiwifruit-growing region of Te Puke in 2010 [12]. This introduction appears to have been a single event, as the Psa population in New Zealand has remained clonal [13]. A reference genome for one of these isolates Psa ICMP 18884 (hereafter referred to as Psa3 V-13) has been fully sequenced [14]. New Zealand strains of Psa3 are distinguished from European, South American and some Chinese isolates by the presence of a unique member of a family of integrative conjugative elements PacICE1-3 [15–17].

Resistance to Psa3 has been observed within an *Actinidia* germplasm collection, including in *A. arguta* [18]. In contrast, in a commercial *A. arguta* orchard, rare Psa infections of *A. arguta* 'HortGem Tahi' and 'HortGem Rua' cultivars produced symptomatic angular necrotic leaf spots; however, the outbreak did not result in a significant loss of orchard productivity [19]. Additionally, limited infection is observed in *A. arguta* seedlings stab-inoculated with Psa, with infection limited to the tissue immediately surrounding the inoculation site [20]. This appears to be related to earlier recognition of Psa3 in *A. arguta* than in *A. chinensis* [21], suggesting that *A. arguta* has a degree of resistance to Psa, which may be conferred by undiscovered resistance genes recognizing Psa3 effectors.

Host range in the *P. syringae* species complex is largely driven by the composition of the effector complement, which consists of at least 68 effector families [22]. Effectors are thus intrinsic to the ability of specialized pathogens within this species complex to cause disease *in planta*. However, an individual *P. syringae* strain carries only a fraction of this pan effector

repertoire; further still, only a subset of these effectors, owing to redundancy within the effector complement, may make an indispensable contribution to virulence in a given host [23,24].

Effector proteins are translocated into host cells via a type III secretion system (T3S) encoded by the *hrp/hrc* gene cluster [25]. The *hrp/hrc* genes are required for the production of the T3S, as Δ*hrcC* deletion mutants cannot deliver effector proteins into host plant cells, thus preventing pathogenicity in host plants [25,26]. Once in host cells, effectors promote bacterial virulence by interacting with host targets to suppress host immunity, allowing the pathogen to invade host tissue, acquire nutrients and cause disease [27–29]. Plant resistance proteins monitor the integrity of, for example, defence signaling cascades, and can detect subversion by bacterial effectors, inducing effector-triggered immunity (ETI), thus restoring plant resistance [28,30,31]. *P. syringae* T3S effectors are termed Hop (Hrp outer protein) or Avr (avirulence) proteins [32]. Avr proteins are a subset of Hop effectors that are recognized by the products of known plant disease resistance genes.

In the majority of *P. syringae* genomes, two groups of effectors are co-located with the *hrp/hrc* gene cluster, forming a tripartite pathogenicity island. These are the conserved effector locus (CEL) and the more variable exchangeable effector locus (EEL) [25]. CEL effectors are required for pathogenesis, demonstrated by strongly reduced pathogenicity and virulence in *P. syringae* ΔCEL strains in host plants [25,26,33]. The EEL has been remodeled extensively between different *P. syringae* pathovars, creating significant genetic variation through mutation, insertion, deletion, and recombination [3]. The EEL from Psa3 ICMP 18884 (non-syntenic compared to other *P. syringae* genomes) contains the effectors *hopQ1a*, *hopD1a*, *avrD1*, *avrB2b*, *hopAB1b*, *hopF4a*, *hopAW1a*, *hopF1e*, *hopAF1b*, *hopD2a*, and *hopF1a*. However, even within the Psa pathovar, the EEL is variable across biovars and strains [15,34].

While previous research has identified that Psa3 CEL (and related) effectors are required for virulence [26], no specific Psa3 avirulence effectors recognized by *Actinidia* spp. have been identified. In this study, a genome sequencing field survey and subsequent effector knockout assays identified four Psa3 avirulence effectors associated with the resistance response to Psa3 in *A. arguta* AA07_03: HopAW1a, AvrRpm1a, HopF1c, and HopZ5a.

## Results

### Psa3 induces the hypersensitive response in *Actinidia arguta*

Previous work showed that *A. arguta* plants were resistant to Psa3, associated with a quantifiable increase in ion leakage due to membrane disruption of the dying cells indicative of a hypersensitive response (HR) [35]. Leaves of *A. arguta* and *A. chinensis* var. *chinensis* spray-infected with Psa3 were visually inspected for macroscopic symptoms at 1 day post inoculation (dpi) (*A. arguta*) or 5 dpi (*A. chinensis* var. *chinensis*). This revealed small dark brown patches each consisting of a few cells, indicative of an HR in *A. arguta*, in contrast to leaves of *A. chinensis* var. *chinensis* (Fig 1A). Accumulated phenolic compounds, characteristic of the HR, were more obvious when the leaves were cleared (Fig 1A). At higher magnification, immunolabelling with an antibody specific for β1-3-glucan revealed callose accumulation in the portion of the cell walls of live cells in direct contact with the dead cells in *A. arguta*, and a lack of cell death, but some callose deposition in *A. chinensis* var. *chinensis* (Fig 1B). Under a fluorescence microscope the dead mesophyll cells were readily visible because of their high concentrations of phenolic compounds following cell wall degradation. Collectively these results in *A. arguta* treated with Psa3 show hypersensitive cell death and a defence response in adjacent cells, hallmarks of ETI.

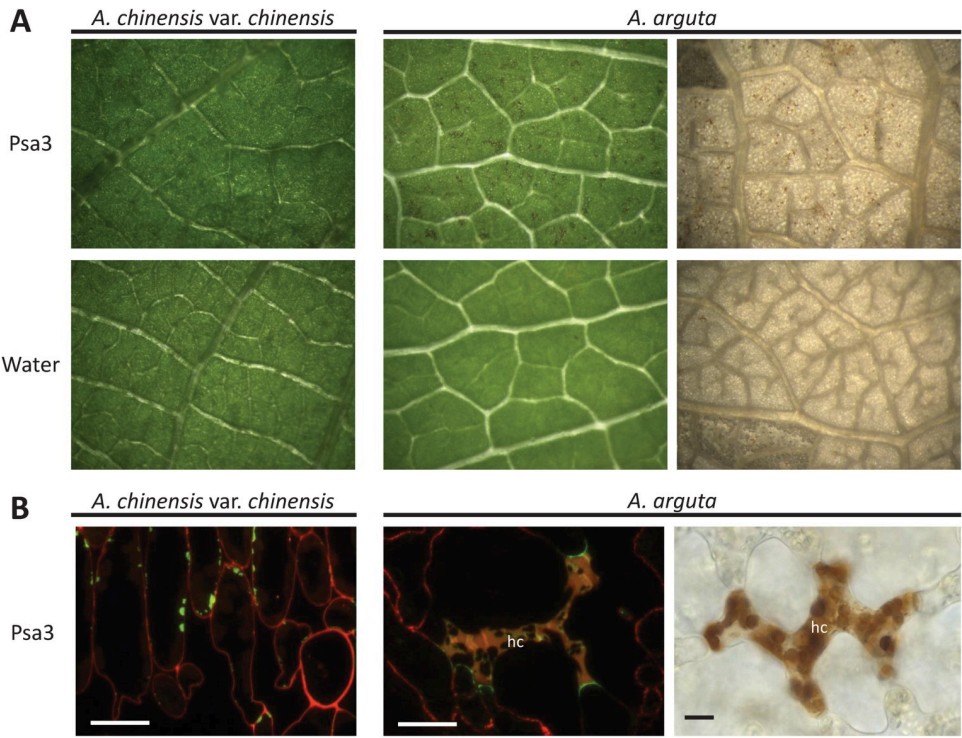

**Fig 1.** ***Pseudomonas syringae* pv. *actinidiae* biovar3 (Psa3) induces the hypersensitive response in *Actinidia arguta*.** *A. arguta* and *A. chinensis* var. *chinensis* leaves displaying symptoms following infection with Psa3 V-13 at 1 day post-infection (dpi) in *A. arguta* and 5 dpi in *A. chinensis* var. *chinensis*. **(A)** Visualization of macroscopically visible localized cell death indicative of a hypersensitive response (HR) in leaves of *A. arguta*, in contrast to *A. chinensis* var. *chinensis*, spray-infected with Psa3 V-13 at $10^8$ cfu/mL or water control (left). *A. arguta* leaves were cleared in acetic acid:ethanol to better visualize brown phenolic compounds indicating cell death (right; brown speckling in the images). Images were viewed under a binocular microscope at 2x magnification, except for the top right image which was at 4x magnification. **(B)** Fluorescence microscopy of Psa3 V-13-infected *A. arguta* and *A. chinensis* var. *chinensis* mesophyll tissue. Callose (β1-3-glucan) is immuno-labelled and fluorescence indicated in green; cell wall pectin is immuno-labeled and fluorescence indicated in red; yellow coloring is accumulation of phenolic compounds in cells showing hypersensitive cell death (hc; left) and loss of cell wall integrity. Bright field microscopy of cleared *A. arguta* leaf in **A** indicates phenolic compound accumulation in cells showing hypersensitive cell death (hc; right) caused by cell wall breakdown. Scale bars represent 10 μm.

## Psa isolates from symptomatic *A. arguta* leaves have a 51 kb deletion in the exchangeable effector locus

During routine surveys of our *Actinidia* spp. germplasm collection at the Te Puke Research Orchard, we observed leaves of *A. arguta* 'HortGem Tahi' with leaf spot disease symptoms. The leaf spots comprised an angular necrotic zone surrounded by a chlorotic halo (Fig 2A). *P. syringae* was isolated from these lesions and confirmed to be Psa3 using qPCR [36,37]. Several of these isolates were sequenced using the Illumina HiSeq platform. Four isolates had a 51 kb deletion in the EEL, with the deletion flanked by Insertion Sequence (IS) 630 DDE endonucleases (pfam00665) and MITEs [38, 39]. The deleted region contained several effectors including *hopAW1a*, *hopF1e*, *hopAF1b*, *hopD2a*, and *hopF1a*, and genes encoding a putative novel non-ribosomal peptide synthase (NRPS) toxin synthesis pathway (Fig 2B). One of the isolates with the 51 kb deletion, Psa3 X-27, was checked by PCR spanning the deletion site (*Psa-X27*; S1 Table) and Sanger sequencing to confirm the deletion (Fig 2C).

Potted plants of *A. arguta* AA07_03 were infected with Psa3 10627 (wild type, WT) and one of the isolates with the 51 kb deletion, Psa3 X-27. Leaves of these Psa3 X-27-infected plants

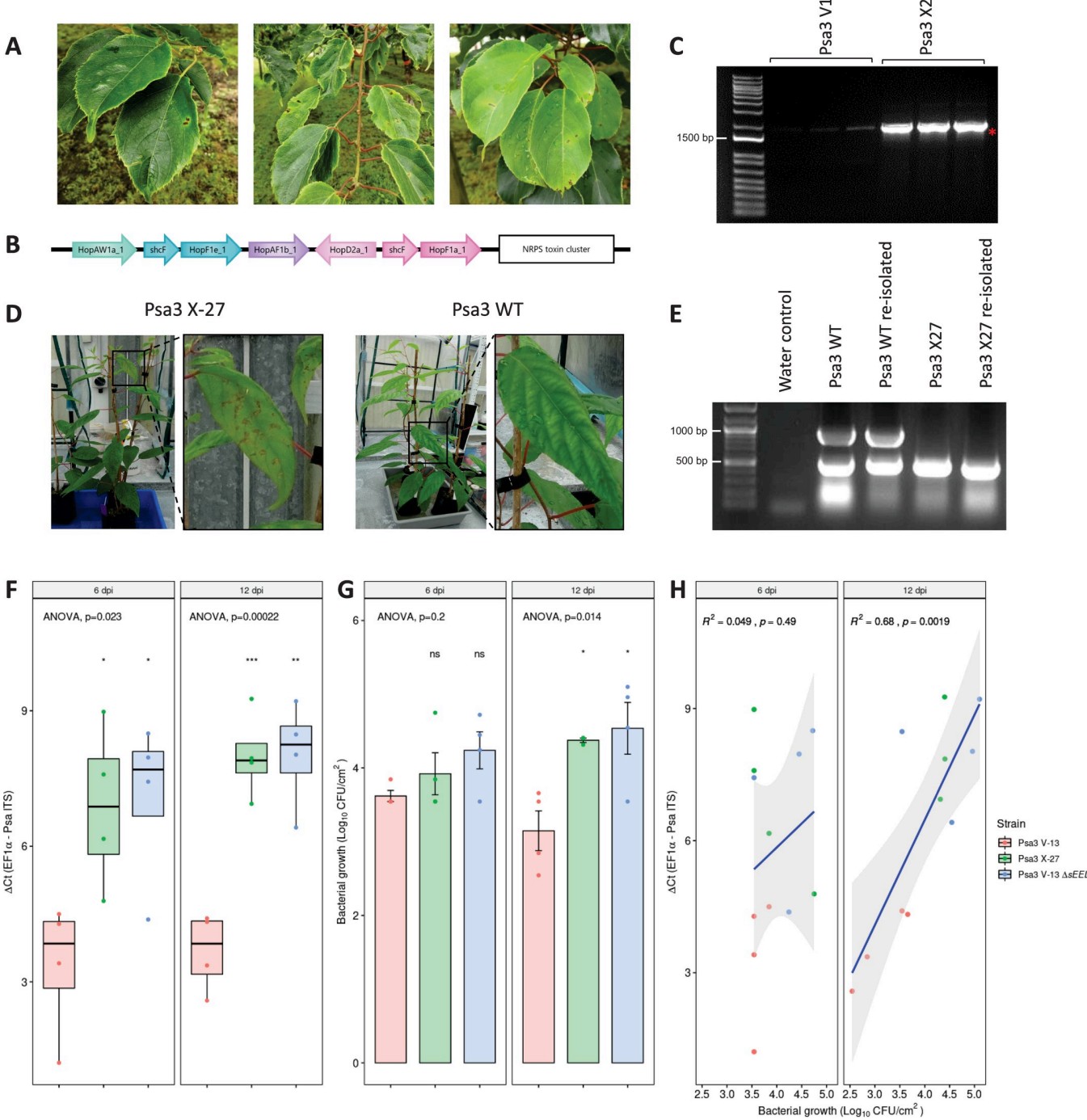

**Fig 2. *Pseudomonas syringae* pv. *actinidiae* biovar3 (Psa3) isolated from symptomatic *Actinidia arguta* plants has a deletion in the exchangeable effector locus that escapes host recognition. (A)** Psa leaf spot symptoms on commercial *A. arguta* 'HortGem Tahi' plants in the Plant & Food Research Te Puke Research Orchard. **(B)** The Psa3 X-27 gene deletion spans the effectors *hopAW1a*, *hopF1e*, *hopAF1b*, *hopD2a*, *hopF1a*, and the non-ribosomal peptide synthase (NRPS) toxin cluster. The Psa3 X-27 gene deletion was identified through whole-genome sequencing on an Illumina HiSeq platform and confirmed by PCR. **(C)** Three colonies of Psa3 ICMP 18884 (V13) or Psa3 X-27 were used as templates for PCR across the deletion boundary *Psa-X27* (1804 bp) and the band indicating deletion (red asterisk) confirmed by Sanger sequencing. The weak non-specific band in Psa V13 samples are present in all samples. DNA marker is 1Kb Plus DNA Ladder from Thermo Fisher (NZ) with 1500 bp band indicated. **(D)** Psa3 X-27 or Psa3 10627 (WT) were sprayed onto potted *A. arguta* AA07_03 plants and photographs of symptoms taken 6 months post-infection. **(E)** Psa3 10627 (WT) and Psa3 X-27 re-isolated from infected leaves and confirmed by multiplex PCR for *Psa-ompP1* (492 bp) and the EEL effector gene *hopF1e* (883 bp). DNA ladder is 100bp DNA Marker from Zymo Research (USA) with 1000 bp and 500 bp bands indicated. **(F-H)** *A. arguta* AA07_03 plantlets were flood-inoculated with Psa3 V-13, Psa3 X-27, and Psa3 V-13 Δ*sEEL* at approximately $10^6$ cfu/mL. Bacterial growth was quantified at 6 and 12 days post-inoculation by qPCR ΔCt analysis **F** and plate count quantification **G**. **(F)**

Box and whisker plots, with black bars representing the median values for the four pseudobiological replicates and whiskers representing the 1.5 inter-quartile range. **(G)** Bar height represents the mean number of $Log_{10}$ cfu/cm$^2$ and error bars represent the standard error of the mean (SEM) between four pseudobiological replicates. **(H)** Regression analysis comparing the two quantification methods (**F** and **G**). The linear regression line is indicated in blue, the grey region indicates a 95% confidence interval, and the r-value represents the correlation coefficient (R$^2$) and its associated p-value. The experiments were repeated three times with similar results. Asterisks indicate the statistically significant difference of Student's *t*-test between the indicated strain and wild-type Psa3 V-13, where p≤.05 (*), p≤.01 (**), p≤.001 (***), and p>.05 (ns; not significant).

had chlorotic halos and necrotic leaf symptoms, in contrast to plants infected with Psa3 (WT), which displayed no visible symptoms (Fig 2D). Psa3 X-27 and Psa3 WT were re-isolated from the spray-infected leaves and verified by PCR. Here, confirmation was achieved by multiplex PCR for EEL locus effector gene *hopF1e* (883 bp) and Psa-*ompP1* primers [40] (492 bp), with both present in the WT but only Psa-*ompP1* present in the original and the re-isolated X-27 (Fig 2E).

## Psa3 X-27 escapes host recognition in *A. arguta* through effector loss

To determine whether it was the Psa3 X-27 multi-effector deletion that allowed this isolate to overcome *A. arguta* resistance, the Psa3 V-13 Δ*sEEL* knockout strain was developed to have the same EEL effector deletion as Psa3 X-27 while retaining the putative NRPS toxin biosynthesis gene cluster. *A. arguta* AA07_03 plantlets were flood-inoculated with Psa3 V-13, Psa3 X-27, and Psa3 V-13 Δ*sEEL* and assessed for *in planta* growth for a single experimental run (Fig 2F and 2G). At 0 dpi, Psa3 V-13, Psa3 X-27, and Psa3 V-13 Δ*sEEL* each had mean bacterial biomass of 5.6, 5.7, and 5.7 $Log_{10}$ cfu/cm$^2$ respectively (S1 Fig). Infected plantlets were sampled at 6 and 12 dpi. Psa3 V-13 triggered resistance in *A. arguta* AA07_03 at 6 and 12 dpi; a 5-fold increase in bacterial biomass was observed for both Psa3 X-27 and Psa3 V-13 Δ*sEEL* relative to Psa3 V-13 using the qPCR approach (Fig 2F). Qualitatively, this same trend was also observed using the plate count method to quantify Psa biomass (Fig 2G). A linear correlation was observed when the dependent variables from the plate count ($Log_{10}$ cfu/cm$^2$) and qPCR (ΔCt) methodologies were plotted against one another as a regression analysis, specifically at 12 dpi (Fig 2H).

At 50 dpi, AA07_03 plantlets inoculated with Psa3 V-13 appeared healthy, with little to no development of disease symptoms (S2 Fig). Conversely, AA07_03 plantlets inoculated with Psa3 X-27 developed leaf yellowing with small, angular, necrotic lesions surrounded by chlorotic halos. Similar disease-like symptoms were observed when AA07_03 was inoculated with Psa3 V-13 Δ*sEEL* (S2 Fig). Quantification of diseased tissue (chlorotic and necrotic tissues) using a PIDIQ pipeline [41] indicated a clear difference between Psa3 V-13-infected versus Psa3 X-27- and Psa3 Δ*sEEL*-infected plants (S3 Fig). Unlike AA07_03, *A. chinensis* var. *chinensis* 'Hort16A' is highly susceptible to Psa3 V-13. At 50 dpi, 'Hort16A' plantlets inoculated with Psa3 V-13 had a high degree of leaf yellowing and large areas of necrosis (S2 Fig). Psa3 X-27 and Psa3 V-13 Δ*sEEL* both produced similar disease symptoms to Psa3 V-13 in 'Hort16A', with widespread necrosis evident (S2 Fig).

## Four candidate avirulence effector loci contribute to Psa3 recognition in *A. arguta*

Knocking out the sEEL locus increased virulence in *A. arguta* AA07_03 quantitatively, but Psa3 X-27 or Psa3 V-13 Δ*sEEL* were not as virulent in *A. arguta* AA07_03 as they were in 'Hort16A' (S2 Fig). This suggested that there may be additional effectors recognized by AA07_03 within the Psa3 V-13 effector complement.

To determine whether additional Psa3 V-13 effectors triggered resistance in *A. arguta*, a library of 21 knockout strains was generated, covering all 30 effectors from Psa3 V-13,

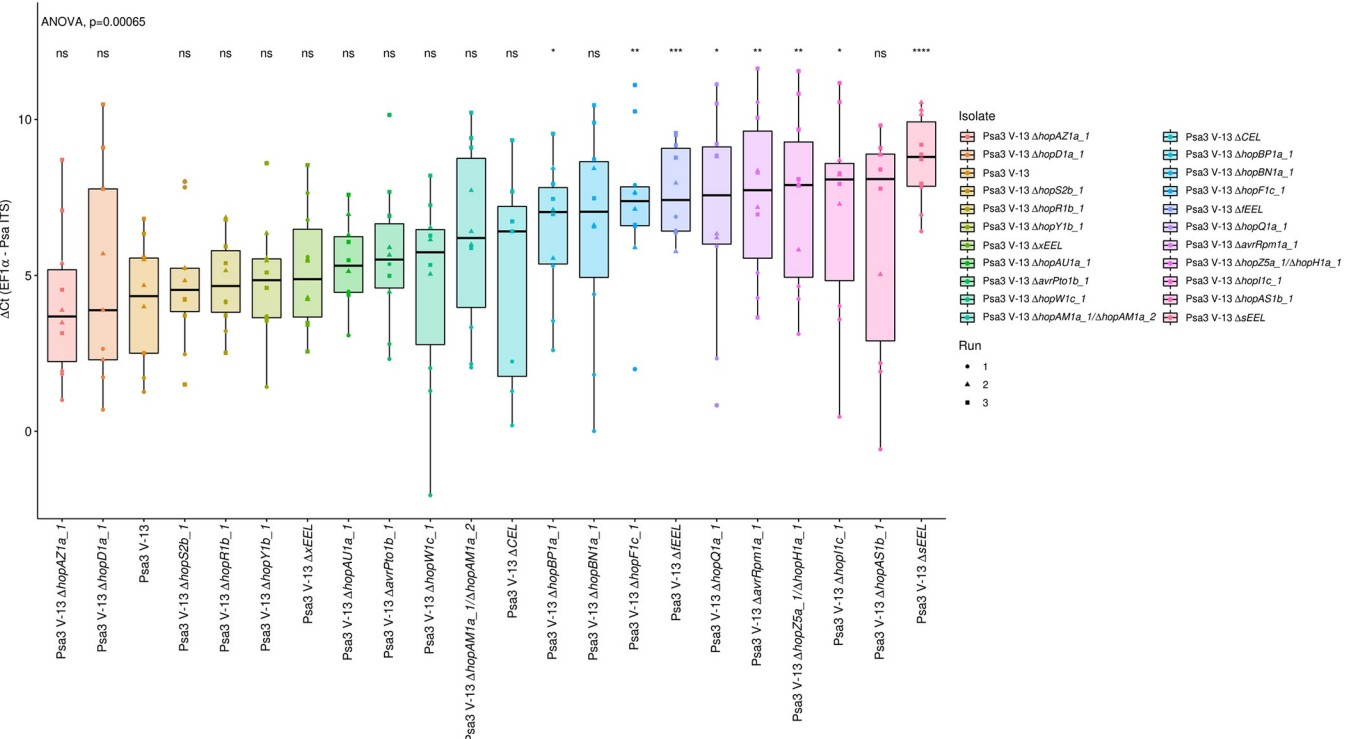

**Fig 3. Pathogenicity assay screen of Psa3 V-13 effector knockout strains in *Actinidia arguta* identifies four avirulence loci.** *A. arguta* AA07_03 kiwifruit plantlets were flood-inoculated at approximately $10^6$ cfu/mL. Psa biomass (*ITS*) was quantified relative to *AaEF1α* using the ΔCt analysis method for three pseudobiological replicates, per strain, per experimental run. Box and whisker plots, with black bars representing the median values and whiskers representing the 1.5 inter-quartile range. Asterisks indicate the statistically significant difference of Student's *t*-test following ANOVA between the indicated strain and wild-type Psa3 V-13, where p ≤.05 (*), p≤.01 (**), p≤.001 (***), p≤.0001 (****), and p>.05 (ns; not significant). This experiment was separately conducted three times (biological replicates) with three batches of independently grown plants and data were stacked to generate the box plots.

consisting of 15 individual effectors, a redundant effector pair (*hopAM1a-1/hopAM1a-2*), effector blocks (*hopZ5a/hopH1a*, CEL, or three different iterations of the EEL–S4 Fig). This library of knockout strains was screened in *A. arguta* AA07_03 plantlets by flood-inoculation and sampled at 12 dpi (Fig 3). qPCR bacterial biomass quantification alone was used for this screen, owing to the large number of strains being assessed for pathogenicity across three independent infection experiments.

Several effector knockout strains achieved significantly more bacterial growth than Psa3 V-13, including Psa3 V-13 Δ*sEEL*, partially escaping recognition in AA07_03 (Fig 3). Additionally, Psa3 V-13 Δ*fEEL*, which encompasses the sEEL alongside additional effectors in the EEL (*avrB2b*, *avrD1*, and *hopF4a*), was also significantly more virulent in AA07_03 than Psa3 V-13 (Fig 3). Conversely, Psa3 V-13 Δ*xEEL*, which encompasses the fEEL effectors alongside additional effectors in the EEL (*hopQ1a* and *hopD1a*), and Psa3 V-13 Δ*CEL* were not significantly different from Psa3 V-13. Psa3 V-13 Δ*hopZ5a*/Δ*hopH1a*, Psa3 V-13 Δ*avrRpm1a* and Psa3 V-13 Δ*hopF1c* also had a significant increase (p ≤ 0.01) in bacterial growth *in planta* relative to Psa3 V-13. The isolates Δ*hopI1c*, Δ*hopBP1a*, and Δ*hopQ1a* had a significant increase (p ≤ 0.05) and Δ*hopBN1a* was not significant overall but was significant in two of the three qPCR runs. These mutants were further tested by plate count methods and were not found to be significantly increased in virulence in AA07_03 compared with Psa3 V-13 (S5 Fig).

Following this screen, candidate avirulence effector knockout strains with significance (p<0.01) were tested by plate count methods. Using the previously described biolistic co-expression assays to measure HR-mediated reporter eclipse in AA07_03 leaves, *hopZ5a* was

identified as the recognized effector in the *hopZ5a*/*hopH1a* effector block (S6 Fig). This assay, described previously, assesses whether biolistic co-delivery of an effector is able to suppress expression of a reporter *GUS* gene due to HR [35,42]. Therefore, only the single *hopZ5a* knockout strain was used for subsequent experiments. Thus, the candidate avirulence-effector knockout strains selected for further analysis were Psa3 V-13 Δ*sEEL*, Psa3 V-13 Δ*fEEL*, Psa3 V-13 Δ*xEEL*, Psa3 V-13 Δ*hopZ5a*, Psa3 V-13 Δ*hopF1c*, and Psa3 V-13 Δ*avrRpm1a*. Psa3 V-13 Δ*hopI1c* was selected to be a negative control in this experiment, as this strain did not display an increase in bacterial growth or escape recognition because of the deletion of the Δ*hopI1* effector gene (S5 Fig). To confirm the candidate avirulence effector knockout strains identified in the qPCR screens, bacterial growth was quantified in AA07_03 using both qPCR and the plate count method (Figs 4A and 4B and S7 and S8). Interestingly, all three of the *EEL* knockout strains had significantly more Psa biomass *in planta*, with a ten-fold increase in bacterial growth relative to Psa3 V-13. Similarly, Psa3 V-13 Δ*hopZ5a*, Psa3 V-13 Δ*hopF1c* and Psa3 V-13 Δ*avrRpm1a* also had significantly more bacterial growth *in planta* relative to Psa3 V-13, with approximately a mean ten-fold increase in bacterial growth. As expected, Psa3 V-13 Δ*hopI1c* was not significantly different from Psa3 V-13.

## sEEL effector HopAW1a triggers resistance in *A. arguta*

Pathogenicity screening of AA07_03 determined that Psa3 V-13 Δ*sEEL* lost at least one avirulence effector (Fig 4A and 4B). To identify which sEEL effector(s) triggers resistance, individual sEEL effectors (*hopAW1a*, *hopD2a*, *hopF1e* and *hopAF1b*) were plasmid-complemented into Psa3 V-13 Δ*sEEL* (S2 Table). Pathogenicity assays were conducted to identify which sEEL effector(s) triggered resistance in *A. arguta* AA07_03.

Plasmid complementation of Psa3 V-13 Δ*sEEL* with *hopAF1b* and *hopD2a* yielded similar amounts of *in planta* bacterial biomass to Psa3 V-13 Δ*sEEL* and these were significantly different from Psa3 V-13 (Fig 4C and 4D). This suggests that neither HopAF1b nor HopD2a trigger resistance to Psa3 V-13 in AA07_03. Interestingly, Psa3 V-13 Δ*sEEL* + *p.hopAW1a* and Psa3 V-13 Δ*sEEL* + *p.hopF1e* showed a decrease in *in planta* bacterial biomass relative to Psa3 V-13 Δ*sEEL*, suggesting that individual plasmid complementation of *hopAW1a* and *hopF1e* partially restored host recognition (Fig 4C and 4D). However, using the qPCR method (Fig 4C), *in planta* bacterial biomass of neither of these strains was fully reduced to the same degree as Psa3 V-13, possibly owing to plasmid loss. If both effectors are required for recognition, they may have an additive effect that is only fully seen in wild-type Psa3 V-13. The plate count quantification (Fig 4D), in contrast, showed neither Psa3 V-13 Δ*sEEL* + *p.hopAW1a* nor Psa3 V-13 Δ*sEEL* + *p.hopF1e* was significantly different from Psa3 V-13, suggesting that both HopAW1a and HopF1e may trigger resistance in AA07_03.

To confirm that HopAW1a and HopF1e are candidate avirulence effectors, segmented effector knockouts within the sEEL were generated to confirm these results (Fig 4E and 4F). Psa3 V-13 Δ*hopAW1a* lacks *hopAW1a* while Psa3 V-13 Δ*tEEL* lacks *hopF1e*, *hopAF1b*, *hopD2a* and *hopF1a*. Pathogenicity assays of AA07_03 demonstrated that Psa3 V-13 Δ*hopAW1a* was significantly different from Psa3 V-13 and similar to Psa3 V-13 Δ*sEEL*. In contrast, Psa3 V-13 Δ*tEEL* was not significantly different from Psa3 V-13. This suggests that the individual deletion of *hopAW1a* is sufficient to partially release host recognition and further suggests that none of the effectors in the tEEL triggers resistance on AA07_03. The plate count data (Fig 4F) results corroborate the qPCR data (Fig 4E) and suggest that HopAW1a is the sole sEEL effector responsible for triggering resistance on AA07_03. Notably, AA07_03 plantlets inoculated with Psa3 Δ*sEEL* complemented with *hopAW1a* was the sole plasmid-complemented line to display a lack of disease symptoms, including leaf yellowing and necrosis (S9 Fig). Additionally, Psa3

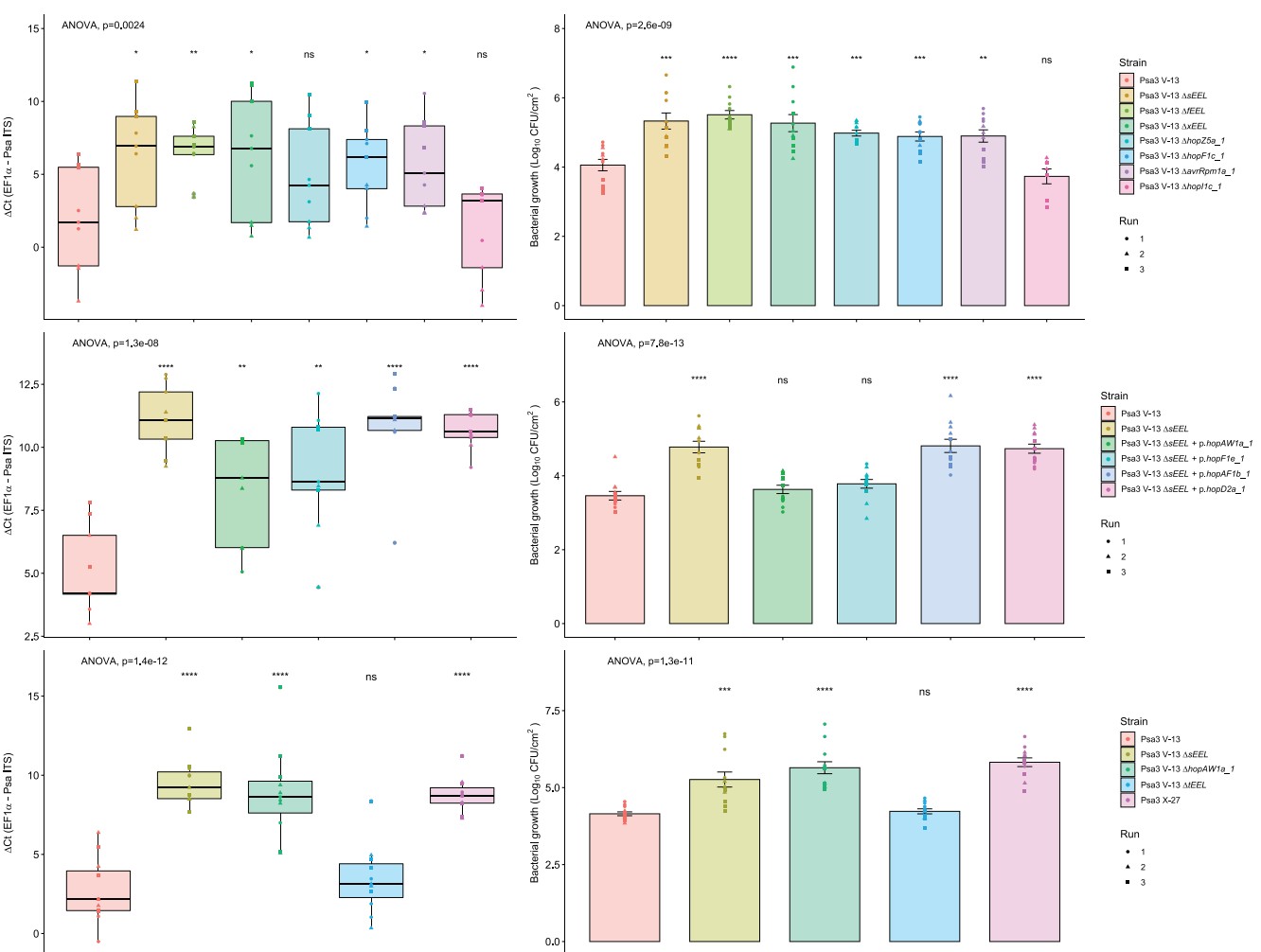

**Fig 4. Pathogenicity assay of Psa3 V-13 effector knockout strains and complementation confirming four effectors' recognition in *Actinidia arguta*.** *A. arguta* AA07_03 kiwifruit plantlets were flood-inoculated at approximately $10^6$ cfu/mL. Bacterial pathogenicity was quantified relative to Psa3 V-13 using the ΔCt analysis method and box and whisker plots, with black bars representing the median values and whiskers representing the 1.5 inter-quartile range in **(A)**, **(C)**, or **(E)**, or plate count quantification with bar height representing the mean $\log_{10}$ cfu/cm² and error bars representing the standard error of the mean (SEM) in **(B)**, **(C)**, or **(F)**, for four pseudobiological replicates, per strain, per experimental run. Bacterial growth was quantified 12 days post-inoculation for selected knockout strains (A) and (B), the plasmid-complemented Δ*sEEL* strains (C) and (D), or the Δ*tEEL* and Δ*hopAW1a* strains (E) and (F). Asterisks indicate the statistically significant difference of Student's *t*-test between the indicated strain and wild-type Psa3 V-13, where p ≤.05 (*), p≤.01 (**), p≤.001 (***), p≤0.0001 (****), and p>.05 (ns; not significant). This experiment was separately conducted three times (biological replicates) with three batches of independently grown plants and data were stacked to generate the box plots and bar graphs shown.

Δ*hopAW1a* produced Psa3 Δ*sEEL*-like disease symptoms while Psa3 Δt*EEL* did not (S9 Fig). Quantification of diseased tissue (chlorotic and necrotic tissues) using the modified PIDIQ pipeline indicated Psa3 Δ*sEEL*-infected plants most closely resembled the Psa3 Δ*hopAW1a*-infected plants, while Psa3 V-13-infected plants resembled Psa3 Δ*tEEL*-infected plants (S10 Fig). These results were further supported by biolistic co-expression assays in AA07_03 leaves, with only *hopAW1a* triggering an HR response and an associated reporter eclipse (see below, S11 Fig).

To confirm that the effector deletions were indeed responsible for the increases in virulence seen in the Psa3 knockout strains, each effector was plasmid-complemented in the knockouts and tested for restoration of avirulence. All four avirulence effectors when complemented on

plasmids were able to restore reduced growth as seen for wildtype Psa3 V-13 carrying empty vector (S12 Fig), and each of the knockout strains with plasmid-complemented HA-tagged effectors were also confirmed for secretion of the effector (S13 Fig).

## Psa3 candidate avirulence effectors trigger a hypersensitive response in *A. arguta*

Psa3 V-13 effectors *hopAW1a*, *hopF1c*, *hopZ5a*, and *avrRpm1a* cloned under a 35S promoter were co-bombarded into kiwifruit leaf tissue with a GUS reporter gene to assess if the proteins they encode triggered the hypersensitive response (HR) in *A. arguta* AA07_03 and *A. chinensis* var. *chinensis* 'Hort16A' leaves (Fig 5A). The effector *hopA1j* from *P. syringae* pv. *syringae* 61 was used as a positive control for HR in this assay [35]. Co-bombardment of candidate avirulence effectors *hopAW1a*, *hopF1c*, *hopZ5a* and *avrRpm1a* all demonstrated a decrease in GUS activity on *A. arguta* AA07_03 in comparison to the control (empty vector), indicating that the proteins they encode triggered a hypersensitive response. Surprisingly, HopF1c expression in 'Hort16A' leaves also produced an HR similar to that in AA07_03, albeit without a significant difference in ion leakage compared with the control. The HR triggered by AvrRpm1a, HopZ5a and HopAW1a appeared to be AA07_03-specific, however. To confirm that the AA07_03-recognized avirulence effectors were triggering ion leakage, effectors *hopAW1a*, *hopZ5a*, *avrRpm1a*, and *hopF1c* (without its truncated chaperone), cloned under a synthetic promoter with a C-terminal HA tag, were delivered by *Pseudomonas fluorescens* (Pfo) Pf0-1 carrying an introduced type III secretion system. All Pfo Pf0-1 strains were confirmed to express HA-tagged plasmid-borne effectors by western blots (S14 Fig). Ion leakage assays using Pfo Pf0-1 indicated that only HopAW1a resulted in an increase in conductivity compared with the empty vector control, along with the control HopA1j (Fig 5B). Owing to the lack of a functional ShcF protein, unsurprisingly, HopF1c did not trigger ion leakage. Surprisingly, neither HopZ5a nor AvrRpm1a were able to trigger ion leakage.

## Cumulative deletion of Psa3 candidate avirulence effectors does not result in added fitness in *A. arguta*

To identify whether Psa3 V-13 avirulence effectors *hopF1c*, *avrRpm1a*, *hopZ5a*, and *hopAW1a* contribute cumulatively towards triggering resistance, all four effectors were successively knocked out of the Psa3 V-13 strain and these multiple-knockout strains were inoculated onto *A. arguta* AA07_03 and *A. chinensis* var. *chinensis* 'Hort16A' plantlets (Fig 6). Psa3 V-13 Δ*hrcC* was used as a negative control, as it lacks the ability to secrete type III effectors into host plant cells and is not virulent in *Actinidia* host plants, including 'Hort16A'.

Interestingly, while Psa3 V-13 is avirulent in AA07_03, the type III secretion-deficient mutant (Psa3 V-13 Δ*hrcC*) grew less than the wild-type, suggesting that while several effectors trigger a strong HR in AA07_03 plants, the retention of effector secretion remains largely beneficial to Psa3 (Fig 6A and 6B). Furthermore, pathogenicity assays in AA07_03 demonstrated that, while Psa3 V-13 Δ*hopF1c*/Δ*hopAW1a* (double), Psa3 V-13 Δ*hopF1c*/Δ*hopAW1a*/Δ*avrRpm1a* (triple), and Psa3 V-13 Δ*hopF1c*/Δ*hopAW1a*/Δ*avrRpm1a*/Δ*hopZ5a* (quadruple) were significantly different from Psa3 V-13, they did not cumulatively increase in growth *in planta* with each successive knockout (Fig 6A and 6B). This finding of reduced fitness in AA07_03 for the multiple knockout strains was largely reflected in 'Hort16A', with the quadruple knockout demonstrating nearly 15-fold less growth compared with Psa3 V-13 (Fig 6A). Furthermore, an *in vitro* growth assay of these multiple-knockout strains showed that there are no significant differences in their *in vitro* growth (S15 Fig). Taken together, the data

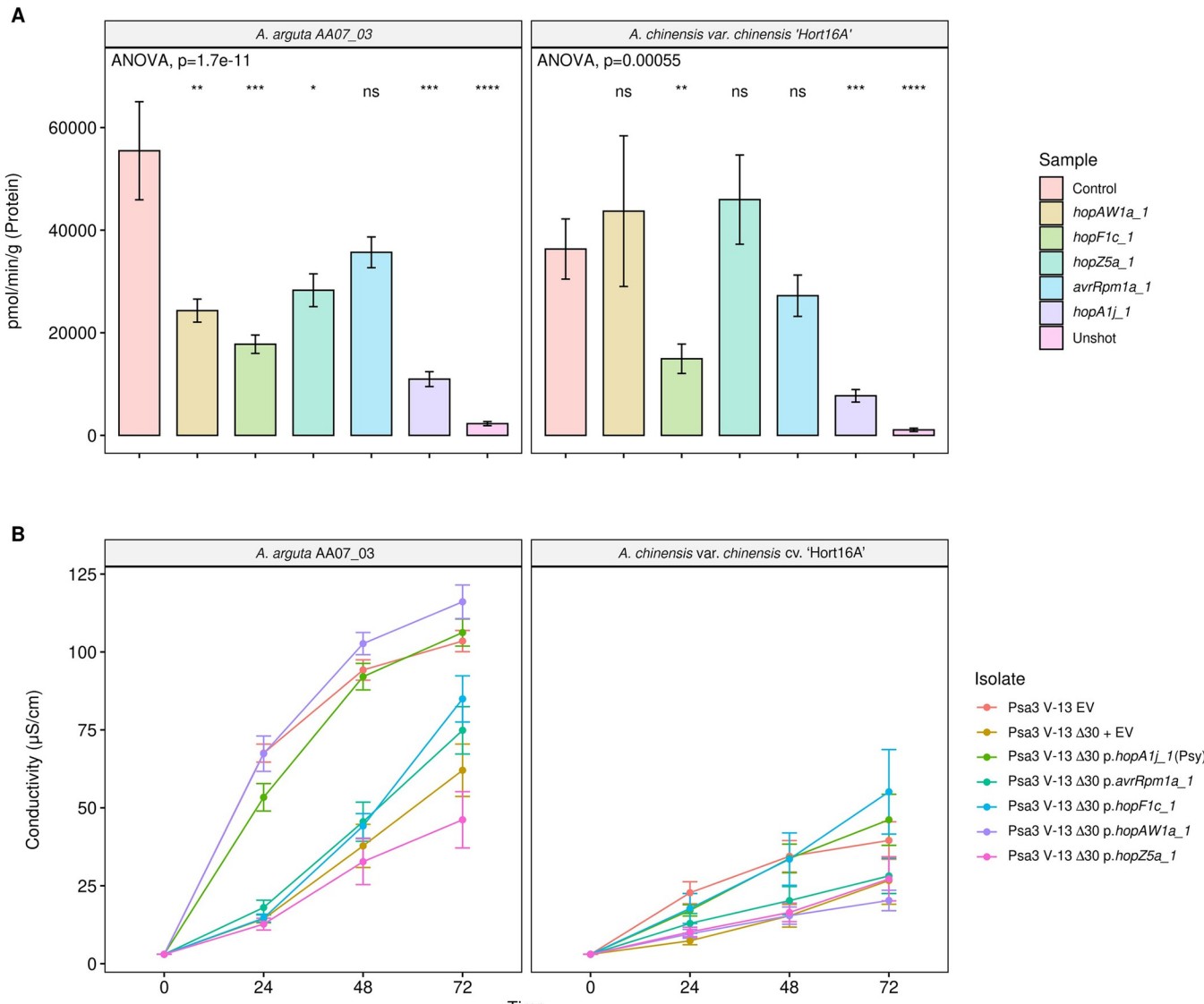

**Fig 5. Reporter eclipse assays demonstrate that HopAW1a, HopZ5a, and AvrRpm1a trigger a host-specific immunity response in *Actinidia arguta* partially supported by ion leakage assays. (A)** Avirulence effectors cloned in binary vector constructs tagged with GFP, or an empty vector (Control), were co-expressed with a β-glucuronidase (GUS) reporter construct using biolistic bombardment and priming in leaves from *A. arguta* AA07_03 or *A. chinensis* var. *chinensis* 'Hort16A' plantlets [35]. The GUS activity was measured 48 hours after DNA bombardment. Error bars represent the standard errors of the means for three independent biological replicates with six technical replicates each (n = 18). HopA1j from *Pseudomonas syringae* pv. *syringae* 61 was used as positive control and un-infiltrated leaf tissue (Unshot) as a negative control. Tukey's HSD indicates treatment groups which are significantly different at α ≤ 0.05 with different letters. **(B)** Leaf discs from *A. arguta* AA07_03 and *A. chinensis* var. *chinensis* 'Hort16A' plantlets were vacuum-infiltrated with *P. fluorescens* PF0-1 wild-type strain (Pfo(WT)) or *P. fluorescens* PF0-1 carrying an artificial type III secretion (Pfo(T3S)), carrying empty vector (EV), or a plasmid-borne type III secreted effector (*hopAW1a, hopZ5a, avrRpm1a* or *hopF1c*, or positive control *hopA1j* from *P. syringae* pv. *syringae* 61) inoculum at ~5 x 10^8 cfu/mL. Electrical conductivity due to HR-associated ion leakage was measured at indicated times over 48 hours. The ion leakage curves are faceted by plant species and stacked for three independent runs of this experiment. Error bars represent the standard errors of the means calculated from the five pseudobiological replicates per experiment (n = 15). Leaf discs from *A. arguta* AA07_03 and *A. chinensis* var. *chinensis* 'Hort16A' plantlets were vacuum-infiltrated with Psa3 inoculum at ~5 x 10^8 cfu/mL. Electrical conductivity due to HR-associated ion leakage was measured at selected time points over 48 hours. The ion leakage curves are faceted by plant species and stacked for three independent runs of this experiment. Error bars represent the standard errors of the means calculated from the five pseudobiological replicates per experiment (n = 15).

suggest that while several Psa3 effectors are recognized in *A. arguta*, the ability to secrete these effectors collectively is beneficial to survival in kiwifruit plants and thus they are unlikely to be lost in succession from a lack of evolutionary selection.

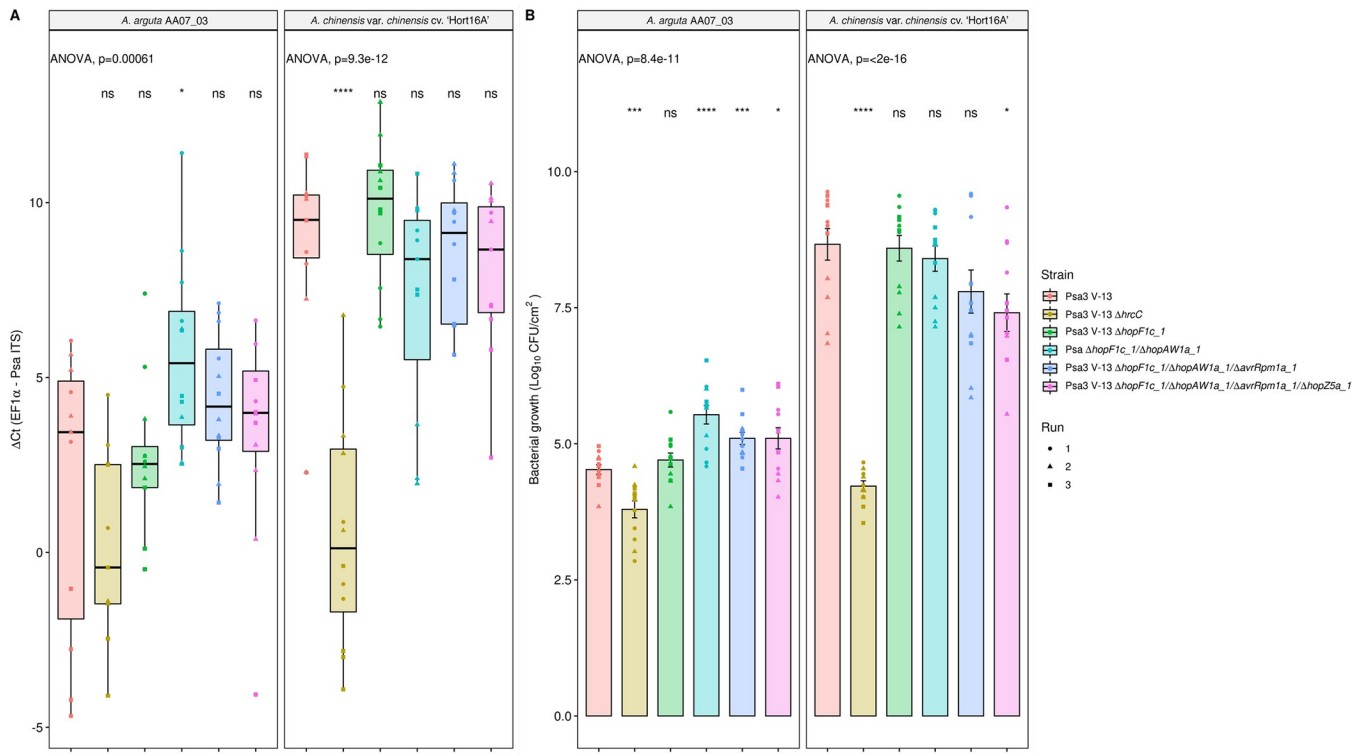

**Fig 6. Pathogenicity assay of Psa3 V-13 multiple avirulence effector knockout strains demonstrates lack of increasing resistance-escape due to a cumulative loss of virulence.** *Actinidia arguta* AA07_03 and *Actinidia chinensis* var. *chinensis* 'Hort16A' kiwifruit plantlets were flood-inoculated at approximately $10^6$ cfu/mL. Bacterial growth was quantified at 12 days post-inoculation using qPCR ΔCt analysis **A** and plate count quantification **B**. The experiment was conducted three times (biological replicates) with three batches of independently grown plants and data were stacked to generate the box plots and bar graphs shown. Asterisks indicate significant differences from ANOVA followed by a *post hoc* Student's *t*-test between the indicated strain and wild-type Psa3 V-13, where p ≤005 (*), p≤.001 (***), p≤.0001 (****), and p>.05 (ns; not significant). **(A)** Box and whisker plots, with black bars representing the median values, whiskers representing the 1.5 inter-quartile range, and black dots indicating outliers. **(B)** Bar height represents the mean number of $Log_{10}$ cfu/cm$^2$ and error bars represents the standard error of the mean (SEM) between four pseudobiological replicates.

## Psa3 avirulence effectors shared by multiple Psa biovars appear to contribute to broad Psa resistance in *A. arguta*

The four Psa3 V-13 effectors we have identified that are recognized in *A. arguta* AA07_03 are also present in the effector complements of the other Psa biovars. At least one avirulence effector is shared for each emergent clade of Psa with *hopAW1a* in Psa5/Psa6, *avrRpm1a* in Psa1/Psa6, and *hopF1c* in Psa2/Psa5 (Fig 7A). Because Psa2 possesses a close orthologue of a truncated effector in Psa3 V-13 (*avrRpm1c*), we checked whether AvrRpm1c was also recognized in AA07_03 and 'Hort16A' leaves. Similar to AvrRpm1a, AvrRpm1c from Psa2 K-28 was also recognized specifically in AA07_03 but not in 'Hort16A' (S16 Fig).

Having examined the effector complement of Psa3 V-13, we next sought to examine whether the presence of these shared avirulence effectors predicted performance of these biovars in *A. arguta* AA07_03. Representative Psa biovar strains were screened in *A. chinensis* var. *chinensis* 'Hort16A' and *A. arguta* AA07_03 to test their virulence (Fig 7B). At 12 dpi, the bacterial growth of Psa1 J-35, Psa2 K-28, and Psa5 in *A. chinensis* var. *chinensis* 'Hort16A' was slightly but significantly lower than that of Psa3 V-13, while that of Psa6 was not significantly different (Fig 7B). Conversely, in *A. arguta* AA07_03, Psa1 J-35 and Psa2 K-28 accumulated in significantly higher amounts than Psa3 V-13 at 12 dpi. Similarly, Psa5 accumulated in slightly higher amounts than Psa3 V-13 at 12 dpi, albeit not significantly. Meanwhile, Psa6

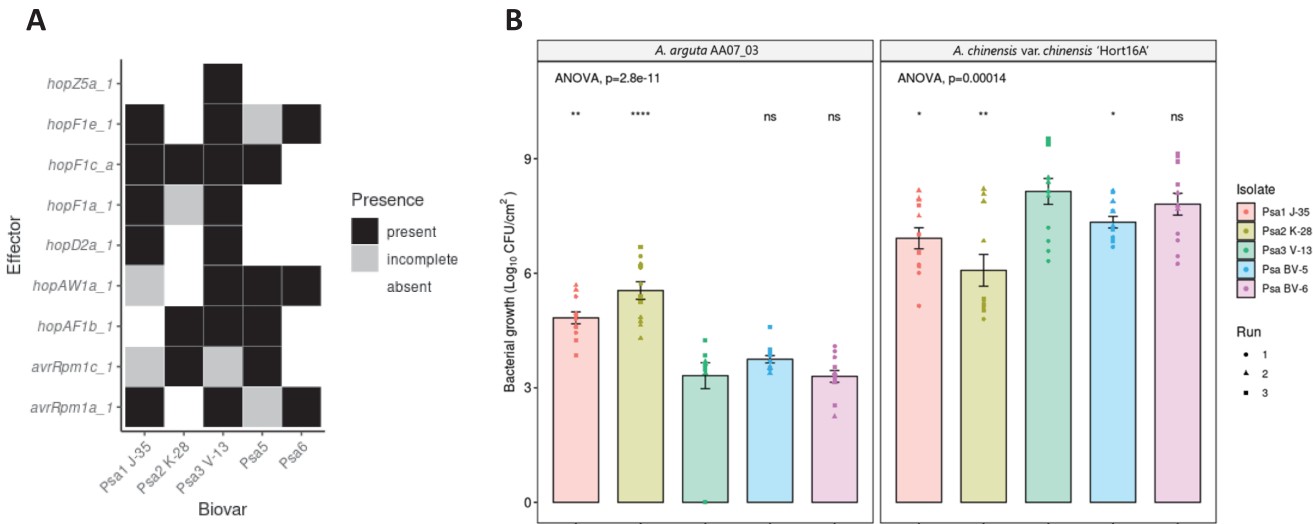

**Fig 7. Pathogenicity assay of *Pseudomonas syringae* pv. *actinidiae* (Psa) biovars in *Actinidia arguta* indicates broad recognition across biovars. (A)** Effectors of interest across strains representative of the Psa biovars. Selected effector repertoires collated from McCann et al. [15] and Sawada et al. [69]. Black indicates when an effector is present; grey indicates when an effector is either truncated, disrupted or incomplete; and white indicates when an effector is absent from a given strain. **(B)** *A. arguta* AA07_03 and *A. chinensis* var. *chinensis* 'Hort16A' kiwifruit plantlets were flood-inoculated at approximately $10^6$ cfu/mL with Psa1 J-35, Psa2 K-28, Psa3 V-13, Psa5 MAFF212057, and Psa6 MAFF212134 strains. Bacterial growth was quantified at 12 days post-inoculation using plate count quantification. The experiment was conducted three times (biological replicates) with three batches of independently grown plants and data were stacked to generate the box plots and bar graphs shown. Asterisks indicate significant differences from ANOVA followed by a *post hoc* Student's *t*-test between the indicated strain and wild-type Psa3 V-13, where p≤.05 (*), p≤.01 (**), p≤.001 (***), p≤.0001 (****), and p>.05 (ns; not significant). Bar height represents the mean number of $Log_{10}$ cfu/cm² and error bars represents the standard error of the mean (SEM) between four pseudobiological replicates.

accumulated *in planta* in amounts similar to those of Psa3 V-13. This relationship between Psa growth in *A. arguta* and *A. chinensis* var. *chinensis* appeared to be inversely correlated. These results taken together suggest a broad recognition, present specifically in *A. arguta*, of a number of shared effectors across the Psa biovars.

## Discussion

Whole genome sequence analysis of a Psa strain isolated from *A. arguta*, Psa3 X-27, identified a 51 kb deletion in the EEL, which included five effectors and an uncharacterized NRPS. This deletion appeared to be the only mutation of significance in these isolates. Flanking the deletion were two DDE IS, DDEg5 and DDEg8, suggesting a relatively facile mechanism for excision of the region via homologous recombination between the DDE loci. Since there are several DDE type IS throughout the Psa3 EEL it also illustrates the ease by which Psa can change its effector profile.

A gene knock out in Psa3 V-13 that deleted the same group of five effectors present in Psa3 X-27 was constructed (Psa3 V-13 *ΔsEEL*). Both these isolates were able to grow to the same extent in AA07_03. This firstly indicated that the deletion of the putative NRPS toxin biosynthesis gene cluster in Psa3 X-27 is not contributing to the increase in *in planta* growth. Several lines of evidence suggest that the increase in bacterial biomass associated with *A. arguta* infections by this strain is due only to the deletion of *hopAW1a*. These include: Psa3 V-13 *ΔsEEL* strain plasmid-complemented with *hopAW1a* demonstrated a decrease in pathogenicity to the same rate as that of Psa3 V-13; the Psa3 V-13 *ΔhopAW1a* individual effector knockout demonstrated an increase in pathogenicity similar to that observed with Psa3 X-27 and Psa3 V-13 *ΔsEEL*; and finally, biolistic expression of HopAW1a in AA07_03 leaves triggered an HR.

To determine whether there were other effectors in Psa3 in addition to HopAW1a that trigger an HR, or whose loss might result in an increase in virulence in *A. arguta*, we generated and screened a library of effector knockouts for their ability to grow in AA07_03. We found that the deletion of Psa3 V-13 effectors *avrRpm1a*, *hopF1c*, and *hopZ5a* increased growth in AA07_03 compared with Psa3 V-13, and biolistic expression of each of these effectors also indicated that they trigger an HR in *A. arguta*. It is interesting that these effectors elicited apparently additive resistance since each knockout relieved avirulence to an extent compared to wildtype. Notably, this approach is unable to identify effectors that non-redundantly participate in virulence but are also recognized in *A. arguta*. The release of ETI from deleting these effectors may result in a failure of effector-triggered susceptibility (ETS). Important Psa effectors that may fall into this category include AvrE1d and HopR1b [26].

The complex interplay of effector complements makes it challenging to dissect the activity of a single effector in isolation. Modular co-expression of *Pseudomonas syringae* pv. *tomato* (Pto) DC3000 effectors has identified multiple instances of effector interplay; for example, the effector AvrPtoB is a suppressor of HopAD1-elicited ETI in *Nicotiana benthamiana* [43]. Similarly, HopI1 can suppress ETI elicited by HopQ1-1 [43]. For the four candidate avirulence effectors identified in this study, AA07_03 must either possess an R protein(s) capable of recognizing more than one effector, or carry multiple R proteins specific for each effector. If these avirulence effectors are collectively recognized by one or more host resistance proteins, the increase in bacterial growth observed in individual effector knockouts may not represent a full escape from host recognition, especially if other avirulence effectors are "unmasked" in the process. There may be further effectors that could be recognized by R proteins that have not been identified in this study owing to suppression of ETI by another effector. Suppression of ETI would prevent a decrease of bacterial biomass in its presence and, therefore, upon deletion we may not detect a change in bacterial biomass.

The pathogenicity assays in this study of *A. chinensis* var. *chinensis* 'Hort16A' and *A. arguta* AA07_03 are among the first to test the virulence of all five described Psa biovars. Psa5 has previously been identified as weakly virulent in the field, while Psa6 has an unknown degree of pathogenicity [9,11]. Similar to Psa3, Psa6 appears to be highly pathogenic in 'Hort16A' but avirulent in AA07_03. We confirmed Psa5 as being less pathogenic in 'Hort16A', similar to the pathogenicities of Psa1 and Psa2 (Fig 7). The strain-specific level of resistance in AA07_03 across the different Psa biovars suggests that there is a complex resistance gene/avirulence effector relationship present [44]. The only partially increased virulence of Psa1 J-35 and Psa2 K-28, relative to that of Psa3 V-13, suggests that these strains may still carry effectors that trigger resistance in AA07_03, including those shared with Psa3 V-13 (Fig 7). Notably, Psa1 carries *avrRpm1a* while Psa2 carries *hopF1c* (and *avrRpm1c*), but Psa 1 and Psa2 may possess other effectors that suppress ETI for these effectors. Nevertheless, the different effector complement of these biovars of Psa suggests a hierarchy of recognition strengths in AA07_03. Namely, HopAW1a recognition confers the strongest growth restriction; Psa1 and Psa2 lacking this effector (as well as *hopZ5a*) have the most growth in AA07_03. HopZ5a/AvrRpm1a/HopF1c confer a similar, lower degree of quantifiable resistance, with effector interplay playing a complex role.

The four avirulence effectors that trigger resistance in AA07_03 can be used to identify cognate resistance proteins and can contribute to effector-assisted breeding in kiwifruit cultivar development programs. Resistance genes that target "Achilles' heel" effectors which are conserved across epidemic strains and several biovars may confer durable, broad-spectrum resistance [45]. For example, *avrRpm1a* is present in Psa1, Psa3 and Psa6, and the closely related *avrRpm1c* is present in Psa2 and Psa5. If these effectors are recognized by the same resistance gene, this might represent a true Achilles' heel for the whole Psa pathovar. Interestingly, testing

of the AvrRpm1c allele from Psa2 K-28 suggested that it is also recognized by AA07_03, possibly by the same resistance protein recognizing AvrRpm1a (S15 Fig). Resistance proteins that target effectors that are variable between strains or biovars are of lower priority for resistance breeding, as they are effective only against a subset of the pathogen population. Unfortunately, *hopZ5a* is unique to the pandemic lineage of Psa3. Similarly, *hopF1c* is absent from Psa1 and Psa6, and *hopAW1a* is absent in Psa1 and Psa2. Of further concern around the utility of resistance against EEL-based effectors, genes located upon the same element could easily be inactivated as a block in a single genetic event, as predicted by Rikkerink et al. [34]. This has already been observed in the field isolate Psa3 X-27, with the deletion of five EEL effectors. This highlights the potential for effector loss under selection pressure from resistant plants in the field. This field-based adaptation underscores the importance of deploying durable resistance genes that ideally target conserved effectors with a virulence requirement, which would impose a fitness cost to a pathogen attempting to escape host recognition.

In contrast, the sequential multiple-effector knockout strategy did not show an additive increase in pathogenicity of Psa3 V-13 in AA07_03. In fact, the quadruple avirulence-effector knockout strain (*ΔhopAW1a/ΔhopF1c/ΔavrRpm1a/ΔhopZ5a*) also had reduced pathogenicity in susceptible *A. chinensis* var. *chinensis* 'Hort16A' plants, supporting the latter conclusion. In addition, the increased pathogenicity of the different Psa biovars in *A. arguta* reflected reduced pathogenicity in *A. chinensis* var. *chinensis*, suggesting a trade-off present in the effector repertoire of Psa. This may be a reason for the high virulence and pandemic spread of Psa3, but not of Psa1 and Psa2, which were earlier emergent diseases of kiwifruit [11,46]. Here it is important to point out that these earlier outbreaks occurred in Korea and Japan where the indigenous *Actinidia* species include *A. arguta* and these biovars therefore presumably evolved partly in the wild *Actinidia* germplasm in Korea/Japan.

A significant observation from this work is that while the deletion of Psa3 effectors recognised by AA07_03 overcomes ETI, *in planta* growth and symptomology is greatly reduced compared to that observed on *A. chinensis* cultivars such as 'Hort16A'. This may reflected a number of things. The remaining effector complement in Psa3 may not be collectively sufficient to efficiently suppress PTI in AA07_03. Alternatively, there may be other preformed defences present in AA07_03 that inhibit the establishment of Psa3 in the apoplast; a failure of ETS reminiscent of non-host resistance [47].

Taken together, these findings highlight a second route to durable resistance: stacking resistance recognition in plants whereby evasion of resistance through loss of multiple effectors will result in cumulative reduced fitness in the plant host. Breeding resistance genes into targeted kiwifruit cultivars is essential for long-term management of Psa. Moreover, breeding *durable* resistance requires an understanding of which pathogen effectors are required for virulence and which trigger resistance in potential hosts. The optimal situation is one where resistance genes target essential effectors, as the loss of an essential effector reduces pathogen fitness *in planta*. Loss of these effectors is, therefore, likely to be selected against. Once identified, resistance genes can be introduced into crops. Traditional breeding can be time-consuming and slow new cultivar development [48]. Alternatively, modern GM technology can efficiently introduce resistance genes without linkage drag of undesirable agronomic traits, to create elite transgenic cultivars [49]. Transgenic crops can also be used to confirm the efficacy of resistance genes before traditional crosses enter pre-commercial field trials, speeding up the cultivar development pipeline. Future research will entail characterizing avirulence effector function, interplay and redundancy to identify which resistance genes are durable breeding targets. Introducing durable Psa resistance that will be effective against the broad spectrum of Psa biovars into future *Actinidia* cultivars will reduce the burden of disease on the horticultural economy and allow a shift towards sustainable production.

## Experimental procedures

### Leaf tissue immunolabelling & microscopy

Pieces of *A. chinensis* var. *chinensis* 'Hort16A' or *A. arguta* AA29_01 leaf, spray-inoculated with Psa3 ICMP 18884 at $10^8$ cfu/mL and harvested at 1–5 days post-infection (dpi), were fixed in 2% paraformaldehyde and 0.1% glutaraldehyde in 0.1M phosphate buffer at pH 7.2 for 1 h under vacuum. Tissue was washed in buffer three times, dehydrated in an ethanol series and embedded in LR White resin (London Resin, Reading, UK) [50]. Sections, 200 nm thick, were cut and dried onto Poly-L-Lysine-coated slides, and left overnight on a hot plate at 45–50˚C. These sections were then immunolabelled [50–52]. Briefly, sections were rinsed in Phosphate-Buffered Saline/Tween (PBS-T), blocked with 0.1% (w/v) bovine serum albumin (Bsa-c, Aurion, Wageningen) in PBS-T for 15 min, and incubated in anti-(1→3)-β-D-glucan antibody (BioSupply, Parkville, Australia) diluted 1:100 in blocking buffer overnight at 4˚C. Sections were then washed in PBS-T, incubated for 1 h in Alexa Fluor 488 goat anti-mouse antibody (Molecular Probes, Eugene, Oreg., USA) diluted 1:600 in PBS, washed in PBS-T, followed by ultrapure water and mounted in Citifluor (Leicester, UK). Sections were viewed on an Olympus Vanox AHTB3 microscope using an interference blue excitation filter set and images collected with a Roper Scientific CoolSnap color digital camera. To highlight the leaf cell walls, sections were either stained with 0.01% (w/v) calcofluor in water (labeling cellulose) or immunolabelled with LM19 (labeling pectin) in a process that followed the initial labelling. The immunolabeling protocol was similar to that described above except that Alexa Fluor 594 goat anti-rat (Molecular Probes) was used as the secondary antibody/fluorchrome combination. The hypersensitive response (HR) was observed by destaining the tissue in acetic acid:ethanol (1:3) for 8 h, washed in 100% ethanol and observed in bright field through the Olympus microscope.

### Field survey & Psa isolation

Samples were taken from leaf spots on vines in the Plant & Food Research Te Puke Research Orchard *Actinidia* germplasm collection. Infected leaves, fruit, bud, shoot and cane samples were taken using secateurs sterilized with 80% ethanol. A 1-cm diameter cork borer was used to punch three leaf discs from each symptomatic leaf. Leaf discs were surface-sterilized in 70% ethanol for 10 s, and washed with sterile MilliQ water in a Falcon tube. For each sample, three leaf discs were placed into an Eppendorf Microcentrifuge Safe-Lock tube (Fisher Scientific, California, United States) with 350 μL sterile 10 mM $MgSO_4$ and three sterile 3.5-mm stainless steel beads. Samples were ground for two runs of 1 min at the maximum speed in a Storm24 Bullet Blender (Next Advance, New York, United States). Tubes were vigorously inverted to resuspend the leaf material pellets between each grinding run. Supernatant (200 μL) was then spread onto lysogeny broth (LB) agar plates [53] supplemented with 12.5 μg/mL nitrofurantoin and 40 μg/mL cephalexin and incubated for 48 h at 22˚C. The bacterial lawn was then re-streaked onto new LB agar plates (supplemented with nitrofurantoin and cephalexin) until single colonies could be isolated.

Quantitative PCR (qPCR) was carried out on an Illumina Eco Real-Time PCR platform (Illumina, Melbourne, Australia), following the protocol outlined by Barrett-Manako and colleagues [36]. Single colonies were tested with Psa-*ITS*, Psa HopZ5-F2/R2 and HopA1-F2/R1 qPCR primers to identify Psa3 strains [37; S1 Table). Samples that amplified in under 35 qPCR cycles were prepared as a 20% (w/v) glycerol stock for long-term storage.

### DNA extraction & sequencing

DNA was purified following the Gentra Puregene protocol for Gram-negative bacteria (Qiagen, Hilden, Germany). Libraries were constructed using the Nextera DNA preparation kit

and sequenced on an Illumina Hi-Seq 2500 platform (paired-end 125 bp reads) (Illumina). Quality control reports for the raw sequencing reads were generated using FastQC [54]. Raw sequencing reads underwent quality and adapter trimming using BBDuk [55] (version 38.62; parameters: ktrim = r, k = 2,1 mink = 11, hdist = 2, minlen = 50, ftm = 5, tpe, tbo, qtrim = r, trimq = 10, minlen = 50, maq = 10). Trimmed reads were mapped to the reference genome Psa ICMP 18884 using the bwa aligner [56] and variants were called using bcftools [57] (version 1.9). Bedtools genomecov was used to generate.bed files of regions with low or no coverage [58]. Bcftools was then used to generate a consensus sequence, masking regions of low or no coverage [57]. Reference genome sequences for the Psa strains used in this study [14] (S3 Table) were obtained from the NCBI GenBank. All downstream analyses were carried out in Geneious [59] (version 10.0.9).

## Microbiological methods

Psa strains used in this study are listed in S2 and S3 Tables. All Psa strains were streaked from glycerol stocks onto LB agar supplemented with appropriate antibiotics; plates were sealed and grown for 48 h at 22˚C. Overnight shaking cultures were grown in LB supplemented with appropriate antibiotics and incubated at 22˚C with 200 rpm shaking. LB agar was supplemented with 12.5 µg/mL nitrofurantoin (Sigma Aldrich, New Zealand) and 40 µg/mL cephalexin (Sigma Aldrich) for Psa selection. To select for Psa strains carrying pK18mobsacB, LB agar was supplemented with 50 µg/mL kanamycin. To counter-select against Psa strains carrying pK18mobsacB, LB agar was supplemented with 12.5 µg/mL nitrofurantoin, 40 µg/mL cephalexin, and 5% sucrose (Merck Millipore, New Zealand). To select for Psa strains carrying pBBR1MCS-5B vectors for effector complementation, LB agar was supplemented with 50 µg/mL gentamicin (Sigma Aldrich).

## Rooted plant inoculations and testing

Experiments were conducted as described previously in Vanneste et al. [60]. Briefly, a bacterial suspension for Psa3 X-27 or Psa3 10627 (WT; clonal isolate related to Psa3 ICMP 18884) [60] was made in water from freshly grown colonies on King's B agar plates [61] and adjusted to ~$10^8$ cfu/mL. Suspensions were sprayed onto the abaxial side of all leaves of three 3- to 4-month-old seedlings of *A. arguta* AA07_03 or *A. chinensis* var. *chinensis* 'Hort16A'. Plants were kept at approximately 20˚C in plastic chambers to maintain the relative humidity. Leaf samples were taken at 14 dpi to re-isolate bacterial DNA for PCR confirmation using *Psa-ITS* and *Psa-ompP1* primers (S1 Table) as described previously [60]. Leaf symptomology photographs were taken at 6 months post-infection.

## Psa3 effector gene knock-out library

Psa3 V-13 was used as the WT for a Psa effector knockout library using the pK18mobsacB-based system. A complete library of 25 Psa3 V-13 effector knockout strains was developed with effectors knocked out either individually, in pairs if homologs were present (*hopAM1a-1*/*hopAM1a-2*) or as a functional group (CEL, EEL various iterations, *hopZ5a*/*hopH1a*, or *hopQ1a*/*hopD1a*) (S2 Table). Effector knockout plasmids were developed for Psa3 V-13 using the methodology established by Kvitko and Collmer [62] and as described in Jayaraman et al. [26]. Briefly, flanking regions 1kb upstream (UP) and 1kb downstream (DN) of the effectors of interest were PCR-amplified with UP-R and DN-F cloning primers carrying an inserted *Xba*I site (S1 Table), digested with *Xba*I restriction enzyme (New England Biolabs/NEB, MA, USA), and ligated to form a 2 kb knockout fragment. This 2 kb fragment was subsequently cloned into the *Eco53k*I restriction enzyme (NEB) site of pK18B-E [26]. The knockout fragment

sequence and quality were verified by sequencing using M13F and M13R primers (Macrogen, South Korea). Psa3 V-13 was transformed with each knockout vector by electroporation (see Plasmid transformation section below). Transformants were plated onto LB agar supplemented with kanamycin to select for strains carrying a genomic insertion of the pK18B-E knockout construct. Resultant colonies were streaked onto LB agar supplemented with 5% sucrose to counter-select against the *sacB* gene in pK18B-E. Resulting colonies were then screened using PCR (check-F/R) primers that amplified outside the knockout region (S1 Table). Successful knockout strains were sub-cultured from 5% sucrose plates onto LB agar supplemented with or without 50 μg/mL kanamycin to confirm plasmid loss and restored kanamycin sensitivity, and the ~2 kb knockout fragment PCR amplicon was sequenced to confirm authenticity (Macrogen, South Korea). The Psa3 Δ*CEL* and Psa3 Δ*hopR1* strains included in the effector knockout strain library were described and characterized earlier [26].

## Plasmid transformations into Psa3

Effector genes were plasmid-complemented back into Psa3 V-13 Δ*sEEL* or individual effector knockout strains following methodology established in Jayaraman et al. [26]. Psa strains were inoculated into 5 mL LB supplemented with appropriate antibiotics and incubated overnight at 20˚C until mid-log phase was reached ($3\times10^8$ cfu/mL). Cultures (2 mL) were collected by centrifugation at 17,000 *g* at 4˚C and washed in cold sterile water multiple times to induce electro-competency according to the previously defined protocol [63]. The final bacterial pellets were resuspended in 100 μL sterile 300 mM sucrose solution, and plasmid DNA added (200–500 ng per reaction). Electro-competent Psa cells were transformed on the Gene Pulser Xcell Electroporation System (Bio-Rad, New Zealand), supplemented with sterile, antibiotic-free LB and incubated at 22˚C for 1 h with 200 rpm shaking, before plating onto LB agar supplemented with gentamicin for plasmid selection and incubated for 48–96 h at 22˚C.

## Pathogenicity assays

*Actinidia* spp. plantlets were obtained from Multiflora Laboratories (Auckland, New Zealand). Plants were grown in 400-mL lidded plastic 'pottles' on half-strength Murashige and Skoog (MS) Agar, with 3–5 plantlets per pottle. Plantlets were grown in a climate-controlled room at 20˚C with a 16 h/8 h light/dark cycle and used within 2–3 months. Plantlets were infected using an *in planta* flooding assay, as established in McActee et al. [64]. Briefly, kiwifruit plantlets were inoculated by flooding with 500 mL Psa inoculum ($\sim 5 \times 10^6$ cfu/mL) for 3 min, and grown in a climate room at 20˚C with a 16 h/8 h light/dark cycle. Un-inoculated plantlets were occasionally checked throughout the experiments for Psa contamination and none was detected.

To quantify bacterial growth of Psa *in planta*, leaf samples were taken at 6 or 12 dpi. A 0.8-cm diameter cork borer was used to punch four leaf discs per replicate, with four pseudo-biological replicates taken per pottle (n = 16), surface-sterilized, and each ground in 350 μL sterile 10 mM MgSO$_4$ with three 3.5-mm stainless steel beads using a Storm24 Bullet Blender (Next Advance, NY, USA). Leaf homogenate stored at -20˚C overnight prior to PDQeX DNA extraction according to a previously described protocol [35].

A serial dilution of leaf homogenate was prepared to quantify cfu/cm$^2$ by the plate count method. A 10-fold dilution series of leaf homogenate in sterile 10 mM MgSO$_4$ was made, to a final dilution of $10^{-5}$ (*A. arguta*) or $10^{-7}$ (*A. chinensis*). Each 10-fold dilution in the dilution series was spot-plated (10 μL) onto LB agar supplemented with appropriate antibiotics. Plates were incubated for 48 h at 20˚C and resultant colonies were counted to calculate the cfu/cm$^2$. To assess disease phenotypes, plantlets were inoculated at $\sim 1 \times 10^8$ cfu/mL and observed at 50

dpi as established by Jayaraman and colleagues [35]. A modified PIDIQ Image-J macro script [41] was used to assess leaf yellowing and browning.

## Quantitative PCR

Real-time quantitative PCR (qPCR) was carried out on an Illumina Eco Real-Time PCR platform, following the protocol outlined in Barrett-Manako et al. [36], with the annealing temperature lowered to 57°C to improve the efficiency of the *EF1α* SN126 L/R primers. The primers used for qPCR are listed in S1 Table.

## *in vitro* growth assay

100 mL LB was inoculated at an $OD_{600}$ of 0.05 from an overnight LB culture, with three replicates per strain. Flasks were shaken on an orbital shaker at room temperature for 24 hours. $OD_{600}$ was measured every 2 hours until the 12 hour timepoint, with a final measurement at 24 hours.

## Ion leakage

*P. fluorescens* (T3S or WT; S4 Table) carrying empty vector or effector constructs were streaked from glycerol stocks onto LB agar plates with antibiotic selection, were grown for 2 days at 22°C, and were restreaked on fresh agar media, and were allowed to grow overnight. Bacteria were then harvested from plates, were resuspended in 10 mM $MgCl_2$, and were diluted to ~$10^8$ cfu/mL. Vacuum-infiltrations were carried out using a pump and glass bell. Leaves were harvested from the tissue culture tubs and were submerged in 30 ml of bacterial inoculum. The vacuum was run until bubbles were rapidly forming. The vacuum valve was then shut and the air slowly let back in. The infiltration was repeated a second time for those leaves not fully infiltrated and any remaining non-infiltrated leaves were removed, as determined by visual examination. For each treatment, leaf discs (6 mm diameter) were harvested from the uniformly vacuum-infiltrated leaf area and were washed in distilled water for 1 h. Six discs were placed in 3 ml of water, and conductivity was measured over 48 h, using a LAQUAtwin EC-33 conductivity meter (Horiba). The standard errors of the means were calculated from five pseudobiological replicates. Data for each timepoint was analyzed by ANOVA followed by a Tukey's HSD post hoc test.

## Reporter eclipse

Freshly expanded leaves of *A. arguta* AA07_03 were co-bombarded with DNA-coated gold particles carrying pRT99-GUS and pICH86988 with the effector of interest, as described in Jayaraman et al. [35].

## Statistical analysis

Statistical analysis was conducted in R [65], and figures were produced using the packages "ggplot2" [66] and "ggpubr" [67]. *Post hoc* statistical tests were conducted using the "ggpubr" and "agricolae" packages [67,68]. The stats_compare_means() function from the "ggpubr" package was used to calculate omnibus one-way analysis of variance (ANOVA) statistics to identify statistically significant differences across all treatment groups [67]. Welch's *t*-test was used to conduct pair-wise, parametric *t*-tests between an indicated strain and a designated reference strain [67]. The HSD.test() function from the "agricolae" package was used to calculate Tukey's Honest Significant Difference [68].

## Supporting information

**S1 Table. Plasmid cloning and confirmation primers used in this study.**
(DOCX)

**S2 Table. Transgenic Psa3 V-13 effector knockout and plasmid-complemented strains.**
(DOCX)

**S3 Table. Wild-type Psa strains.** All wild-type Psa strains were sourced from the International Collections of Micro-organisms from Plants (ICMP) or the National Agriculture and Food Research Organization (NARO); (designated MAFF).
(DOCX)

**S4 Table. *Pseudomonas fluorescens* plasmid-complemented strains used in this study.**
(DOCX)

**S1 Fig. Plate count quantification of bacterial growth at 0 days post-inoculation.** *A. arguta* AA07_03 plantlets were flood-inoculated with Psa3 V-13, Psa3 X-27, and Psa3 V-13 Δ*sEEL* at approximately $10^6$ cfu/mL. Bar height represents the mean number of $Log_{10}$ cfu/cm$^2$ and error bars represent the standard error of the mean (SEM) between four pseudobiological replicates.
(PDF)

**S2 Fig. Symptom development of Psa3 V-13, Psa3 X-27, and Psa3 V-13 Δ*sEEL* in *Actinidia arguta* and *A. chinensis* var. *chinensis*.** *A. arguta* AA07_03 kiwifruit plantlets were flood-inoculated at approximately $10^7$ cfu/mL. Photographs of symptom development in representative pottles were taken at 50 days post-infection.
(PDF)

**S3 Fig. Quantification of symptom development of Psa3 V-13, Psa3 X-27, and Psa3 V-13 Δ*sEEL* in *Actinidia arguta* and *A. chinensis* var. *chinensis*.** A modified PIDIQ image-based analysis of leaf yellowing and browning, expressed as a normalized arcsine-transformed percentage for symptomology photographs taken at 50 days post-infection (S2 Fig). Methodology adapted and modified from that in Laflamme, Dillon [70].
(PDF)

**S4 Fig. The non-canonical extended exchangeable effector locus (xEEL) encompassing the full EEL (fEEL), short EEL (sEEL), and tiny EEL (tEEL) loci.** Schematic of the effectors comprising the xEEL (I-V; *hopQ1a –hopF1a*), fEEL (II-V; *avrD1 –hopF1a*), sEEL (III-V; *hopAW1a –hopF1a*), and tEEL (IV-V; *hopF1e –hopF1a*) loci in Psa3 V-13 ICMP 18884 strain are indicated. Potential recombination sites are indicated: Miniature Inverted Repeat Transposable Element (MITE; grey diamonds), DDE terminal inverted repeats (white diamonds).
(PDF)

**S5 Fig. Pathogenicity assay of Psa3 V-13 selected effector knockout strains in *Actinidia arguta* AA07_03 confirming lack of contribution towards avirulence.** *A. arguta* AA07_03 kiwifruit plantlets were flood-inoculated at approximately $10^6$ cfu/mL. Bacterial pathogenicity was quantified at 12 days post-inoculation relative to Psa3 V-13 using plate count quantification for four pseudobiological replicates, per strain, per experimental run and error bars represent the standard error of the mean (SEM). Asterisks indicate the statistically significant difference of Student's *t*-test between the indicated strain and wild-type Psa3 V-13, where p≤.001 (****), and p>.05 (ns; not significant). This experiment was separately conducted twice (biological replicates) with two batches of independently grown plants and data were

stacked to generate the bar graphs shown.
(PDF)

**S6 Fig. Biolistic transformation reporter eclipse assay demonstrates that HopZ5a, and not HopH1a, triggers a host-specific immunity response in *Actinidia arguta*.** Avirulence effectors cloned into binary vector constructs tagged with GFP, or an empty vector (Control), were co-expressed with a β-glucuronidase (GUS) reporter construct using biolistic bombardment and priming in leaves from *A. arguta* AA07_03 plantlets [35]. The GUS activity was measured 48 hours after DNA bombardment. Error bars represent the standard errors of the means for three independent biological replicates with six technical replicates each (n = 18). HopI1c was used as the positive control and un-infiltrated leaf tissue (Unshot) as the negative control. Tukey's HSD indicates treatment groups that are significantly different at $\alpha \leq 0.1$ with different letters.
(PDF)

**S7 Fig. qPCR-based pathogenicity assay of Psa3 V-13 selected effector knockout strains in *Actinidia arguta* confirming recognition of four avirulence loci.** *A. arguta* AA07_03 kiwifruit plantlets were flood-inoculated at approximately $10^6$ cfu/mL. Bacterial pathogenicity was quantified relative to Psa3 V-13 using the ΔCt analysis method for four pseudobiological replicates, per strain, per experimental run. Data are presented as box and whisker plots, with black bars representing the median values and whiskers representing the 1.5 inter-quartile range. The data have been faceted by experimental run. Asterisks indicate the statistically significant difference of Student's *t*-test between the indicated strain and wild-type Psa3 V-13, where p $\leq$.05 (*), p$\leq$.01 (**), p$\leq$.001 (***), and p$>$.05 (ns; not significant). These three experiments (biological replications) were separately conducted with three batches of independently grown plants.
(PDF)

**S8 Fig. Agarose plate-based pathogenicity assay of Psa3 V-13-selected effector knockout strains in *Actinidia arguta* confirming recognition of four avirulence loci.** *A. arguta* AA07_03 kiwifruit plantlets were flood-inoculated at approximately $10^6$ cfu/mL. Bacterial pathogenicity was quantified relative to Psa3 V-13 using plate count quantification for four pseudobiological replicates, per strain, per experimental run. The data have been faceted by experimental run. Asterisks indicate the statistically significant difference of Student's *t*-test between the indicated strain and wild-type Psa3 V-13, where p $\leq$.05 (*), p$\leq$.01 (**), p$\leq$.001 (***), and p$>$.05 (ns; not significant). These three experiments (biological replications) were separately conducted with three batches of independently grown plants.
(PDF)

**S9 Fig. Symptom development of Psa3 V-13 *ΔsEEL* strains complemented with plasmids carrying individual sEEL effectors and Psa3 V-13 *ΔtEEL* and *ΔhopAW1a* strains in *Actinidia arguta*.** *A. arguta* AA07_03 kiwifruit plantlets were flood-inoculated at approximately $10^7$ cfu/mL. Photographs of symptom development with representative pottles were taken at 50 days post-infection.
(PDF)

**S10 Fig. Quantification of symptom development of Psa3 V-13 *ΔsEEL* strains complemented with plasmids carrying individual sEEL effectors and Psa3 V-13 *ΔtEEL* and *ΔhopAW1a* strains in *Actinidia arguta*.** A modified PIDIQ image-based analysis of leaf yellowing and browning, expressed as a normalized arcsine-transformed percentage for symptomology photographs taken at 50 days post-infection (S9 Fig). Methodology adapted and

modified from that of Laflamme, Dillon [70].
(PDF)

**S11 Fig. Biolistic transformation reporter eclipse assays demonstrate that HopAW1a is the sole sEEL effector triggering a hypersensitive response in *Actinidia arguta* AA07_03.** *sEEL* effectors in cloned binary vector constructs tagged with GFP, or an empty vector (Control), were co-expressed with a β-glucuronidase (GUS) reporter construct using biolistic bombardment and priming in leaves from *A. arguta* AA07_03 plantlets [35]. The GUS activity was measured 48 hours after DNA bombardment. Error bars represent the standard errors of the means for three independent biological replicates with six technical replicates each (n = 18). Un-infiltrated leaf tissue (Unshot) was used as a negative control. Tukey's HSD indicates treatment groups which are significantly different at $\alpha \leq 0.1$ with different letters.
(PDF)

**S12 Fig. Pathogenicity assay of plasmid-complemented Psa3 V-13 effector knockout strains confirms four effectors' recognition in *Actinidia arguta*.** *A. arguta* AA07_03 kiwifruit plantlets were flood-inoculated at approximately $10^6$ cfu/mL. Bacterial pathogenicity was quantified relative to Psa3 V-13 using plate count quantification. Bar height represents the mean $\log_{10}$ cfu/$cm^2$ and error bars represent the standard error of the mean (SEM).
(PDF)

**S13 Fig. Secretion of *Pseudomonas syringae* pv. *actinidiae* biovar 3 (Psa3) V-13 putative avirulence effectors by plasmid complementation in Psa3 knockout strains during expression *in vitro*.** Wildtype Psa3 V-13 carrying empty vector (+ EV), or Psa3 V-13 avirulence effector knockout strains carrying the plasmid-borne type III secreted effector proteins tagged with $6 \times HA$ (complemented strains) were diluted to 5 x $10^8$ cfu/mL in *hrp*-inducing liquid medium, cells pelleted at 6 hr post-inoculation by centrifugation at 12000 *g*, and supernatant boiled in 1x Laemmli buffer, and western blots conducted using α-HA antibody. Yellow asterisks indicate expected sizes for each tagged protein band. HopF1c is cloned and expressed with its preceding chaperone, ShcF.
(PDF)

**S14 Fig. Secretion of *Pseudomonas syringae* pv. *actinidiae* biovar 3 (Psa3) V-13 putative avirulence effectors by plasmid complementation in *P. fluorescens* Pf0-1 carrying an artificial type III secretion system (Pfo +T3S) during expression *in vitro*.** Pfo(+T3S) carrying empty vector (+ EV), or carrying the plasmid-borne type III secreted effectors from Psa3 V-13 tagged with $6 \times HA$ were diluted to 5 x $10^8$ cfu/mL in *hrp*-inducing liquid medium, cells pelleted at 6 hr post-inoculation by centrifugation at 12000 *g*, and supernatant boiled in 1x Laemmli buffer, and western blots conducted using α-HA antibody. Yellow asterisks indicate expected sizes for each tagged protein band. HopF1c is cloned and expressed with its preceding chaperone, ShcF.
(PDF)

**S15 Fig. *In vitro* growth assay of Psa3 V-13 multiple avirulence effector knockout strains.** Points represents the mean $OD_{600}$ and error bars represent the standard error of the mean (SEM) for three independent biological replicates.
(PDF)

**S16 Fig. Biolistic transformation reporter eclipse assays demonstrate that AvrRpm1c from Psa2 K-28 triggers a host-specific immunity response in *Actinidia arguta* AA07_03.** Effectors in cloned binary vector constructs tagged with Green Fluorescent Protein (GFP), or an empty vector (Control), were co-expressed with a β-glucuronidase (GUS) reporter construct

using biolistic bombardment and priming in leaves from *A. arguta* AA07_03 plantlets [35]. The GUS activity was measured 48 hours after DNA bombardment. Error bars represent the standard errors of the means for three independent biological replicates with six technical replicates each (n = 18). HopA1 from *Pseudomonas syringae* pv. syringae 61 was used as the positive control and un-infiltrated leaf tissue (Unshot) as the negative control. Tukey's HSD indicates treatment groups which are significantly different at $\alpha \leq 0.1$ with different letters. (PDF)

## Acknowledgments

We would like to thank Dr Jo Bowen (PFR), Dr Erik Rikkerink (PFR), and Prof. Andrew Allan (PFR) for critical reading of this manuscript.

## Author Contributions

**Conceptualization:** Lauren M. Hemara, Jay Jayaraman, Mirco Montefiori, Cyril Brendolise, Matthew D. Templeton.

**Data curation:** Lauren M. Hemara.

**Formal analysis:** Lauren M. Hemara, Jay Jayaraman, Carl H. Mesarich, Matthew D. Templeton.

**Funding acquisition:** Jay Jayaraman, Matthew D. Templeton.

**Investigation:** Lauren M. Hemara, Jay Jayaraman, Paul W. Sutherland, Mirco Montefiori, Saadiah Arshed, Abhishek Chatterjee, Ronan Chen, Mark T. Andersen, Carl H. Mesarich, Otto van der Linden, Minsoo Yoon, Magan M. Schipper, Joel L. Vanneste, Cyril Brendolise.

**Methodology:** Lauren M. Hemara, Jay Jayaraman, Mirco Montefiori.

**Project administration:** Matthew D. Templeton.

**Supervision:** Jay Jayaraman, Matthew D. Templeton.

**Validation:** Jay Jayaraman.

**Writing – original draft:** Lauren M. Hemara, Jay Jayaraman, Matthew D. Templeton.

**Writing – review & editing:** Lauren M. Hemara, Jay Jayaraman, Matthew D. Templeton.

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
