## [Decision Letter · Decision Letter 0]

24 Jan 2022

Dear Associate Professor Templeton,

Thank you very much for submitting your manuscript "Effector loss drives adaptation of Pseudomonas syringae pv. actinidiae biovar 3 to the non-host Actinidia arguta" for consideration at PLOS Pathogens. As with all papers reviewed by the journal, your manuscript was reviewed by members of the editorial board and by several independent reviewers. In light of the reviews (below this email), we would like to invite the resubmission of a significantly-revised version that takes into account the reviewers' comments.

When reviewing reviewer comments please also address:

More nuanced explanation for avr status of effectors and effectors working together. For example, loss of an effector causing reduced growth on a plant could be due to loss of the ability of an effector to suppress recognition of another. Include both ETI and failed ETS as a framework for compatibility. Shorter the discussion and highlight major findings and implications more succinctly.

We cannot make any decision about publication until we have seen the revised manuscript and your response to the reviewers' comments. Your revised manuscript is also likely to be sent to reviewers for further evaluation.

Sincerely,

Gitta Coaker, PhD

Guest Editor

PLOS Pathogens

Bart Thomma

Section Editor

PLOS Pathogens

Kasturi Haldar

Editor-in-Chief

PLOS Pathogens

orcid.org/0000-0001-5065-158X

Michael Malim

Editor-in-Chief

PLOS Pathogens

orcid.org/0000-0002-7699-2064

When reviewing reviewer comments please also address:

More nuanced explanation for avr status of effectors and effectors working together. For example, loss of an effector causing reduced growth on a plant could be due to loss of the ability of an effector to suppress recognition of another. Include both ETI and failed ETS as a framework for compatibility. Shorter the discussion and highlight major findings and implications more succinctly.

Reviewer's Responses to Questions

**Part I - Summary**

Reviewer #1: In this manuscript, Hamara et al assessed the ability of several strains and effector mutants thereof to cause disease and grow in kiwi and kiwiberry cultivars. They also used a particle bombardment assay to determine the ability of several effectors to be recognized. The strength of this manuscript is in its description of how they might be involved in host restriction/recognition. Many mutants were made and this allowed the authors to infer the function of several effectors. The weakness is that although hosts are compared, we don’t learn about whether resistance is single or multigenic and we don’t get much new insight about these effectors other than they may be triggering resistance. Furthermore, sometimes alternative explanations of the data were not made. For example, the loss of an effector causing reduced growth on a plant could be due to loss of the ability of an effector to suppress recognition of another effector). The analysis of the data limited the ability of the reader to assess how the mutants behaved relative to eachother (not just relative to a WT strain). A non-specialist may likely have difficulty following the results presented, owing to the way the work is presented and the lack of clear explanations of some of the assays. The discussion is long and often does not focus on the putting the major findings into context in a succinct way.

Reviewer #2: The present manuscript submitted by Hemara et al describes a very interesting and useful study that certain T3Es in Psa3, including AvrRpm1a, HopF1c, HopZ5a and HopAW1a, might be recognized by Actinidia argute, but not by its relative A. chinensis, leading to species-specific resistance. The authors convincingly showed certain T3Es responsible for the resistance by screening of a complete library of Psa3 effector knockout strains, which could be further supported by the isolation of a natural EEL-mutant, which contains a 49 kb deletion and causes disease in A. argute. The experiments were well designed and performed with different setups from this group (Jayaraman et al., 2020; 2021), and it is easy to follow the context. Though the authors discussed in line 448-454, it seems A. argute is very resistant to Psa3, regarding the huge difference in bacterial numbers compared to A. chinensis, raising the question of contribution of ETI in A. argute as well as the resolution of the assays performed. The authors are suggested to address the questions below:

Reviewer #3: The authors have conducted a thorough series of genetic experiments to identify hopAW1a_1 as the genetic basis of Psa3 host rage expansion onto A. arguta based on a Psa3 natural genetic variant with a large deletion in its EEL region. In addition to hopAW1a_1, three other effectors are identified that contribute to decreased bacterial load in A. arguta. The group has invested extensively in the tools and techniques to work with this challenging host and the genetics are sound. With the exception of hopAW1a_1, I don't think I would have been so quick to assign avr status to the other effectors based on their quantitative contributions to load and compatibility but I imagine further study will resolve that. In addition, the authors propose lack of ETI as sufficient for compatibility when plainly that is insufficient. Using the zig zag model as a framework failed ETS is likely a factor here. The four combo mutant supports that interpretation.

Reviewer #4: The authors conducted a genome sequencing field survey and identified a strain of Psa3 (P. syringae pv. actinidiae biovar 3) that cause disease on the normally Psa3 resistant plant Actinidia arguta. This strain possesses a deletion of the exchangeable effector locus (EEL) which the authors demonstrate contributes to the avirulence of Psa3 on P. arguta, in particular HopAW1 from this locus. These are the first Psa3 avirulence genes to be identified in Actinidae. Further screening identified 3 additional effectors that contribute to avirulence. Interestingly, cumulative KOs of these effectors did not increase bacterial growth, likely due to their contribution to virulence. This would suggest a robust immune strategy that recognizes multiple Psa3 effectors that cannot readily be lost without compromising virulence. Overall this a very interesting study that has been well executed, but I do have some comments that should be addressed.

**Part II – Major Issues: Key Experiments Required for Acceptance**

Reviewer #1: (No Response)

Reviewer #2: (1) Psa3 does not grow too much in A. argute as revealed by low bacterial titer, compared to the same inoculation condition in A. chinensis var. chinensis. The difference of bacterial numbers between Psa3 WT and Psa3 △hrcC is low (Figure 6), while there is no Psa3 mutant that shows reduced virulence (Figure 3), even in Psa3 △CEL which was proved to be essential in bacterial pathogenicity in A. chinensis (Jayaraman et al., 2020) and in other pathosystem. This could be due to, not limited to, more robust PTI responses or other specialized secondary metabolites in A. argute. In Figure1, the authors are suggested to investigate transcriptional changes of some defense-related genes, to support specific activation of ETI. There is possibility that A. argute could restrict Psa3 growth with more robust PTI responses, please include Psa3 △hrcC as well.

(2) Figure 2F and 2G, the authors are suggested to include data from 0 dpi; from Figure 2G it seems the Psa3 X-27 and Psa3 V-13 △sEEL did not grow, while there is decrease of bacterial titer in Psa3 V-13 WT. Adding data from 0 dpi could help provide a clear picture.

(3) For the assays using Pfo-delivered protein expression, the authors are suggested to provide protein accumulation data supporting that the missing observations of conductivity changes was not because of the missing protein accumulation (Figure 5).

Reviewer #3: No major issues noted

Reviewer #4: Figure 6B: The decrease in growth observed for the polymutants on the susceptible variety could be explained by a general decrease in bacterial fitness. The author’s should measure the growth of these mutants in vitro compared to the wild-type strain. Decreases in fitness could also explain why an additive increase in virulence was not observed when multiple Avr genes were knocked out.

The authors' complemented the sEEL mutant with hopAW1a (Fig 3D) but did not complement the knock-outs of the other 3 identified avirulence genes (HopZ5a, HopF1c and AvrRpm1a; see Fig 3B). The authors' should complement these 3 mutants to show that resistance is restored. If effectors are epitope tagged, western blots should be conducted to confirm expression (including sEEL+hopAW1a).

Figure 3: Should HopF2 be HopF1c? If so, why does the HopF1c behave differently in Fig 6B?

**Part III – Minor Issues: Editorial and Data Presentation Modifications**

Reviewer #1: Figure 1 comments

It is difficult to assess this data because of the different timing used and the lack of quantification. I suggest showing the Psa3 infection for A. chinensis at 1 day and also show the cleared tissue. Presumably, this analysis was done in multiple leaves on different plants (it’s not specified how many), so the data can be quantified in some way- counting brown spots or patches of brown leaves/area or similar. The callose experiments needs a water control, although I’m not sure this measurement is that informative without quantification of some kind. Please add a size bar to part A of the figure.

Fig 2, How many days until plants were photographed? How many plants looked like the picture shown?

For panel C, there is a faint band for V13. This is never discussed in the main text – why was it used, do we expect to see a band? Or is this supposed to be V13delsEEL?

p.8 line 165 The authors say…The bacteria were not as virulent in A. arguta AA09_3 as they were in “Hort16A’ – here the authors should refer to Fig S1. It might be better to say the disease was more severe on Hort16A, since the authors did not measure the bacterial growth, which I usually consider important for assessing virulence.

For the general reader, it would be a good idea to explain the biolistic assay briefly in the text the first time that you do it. I assumed this was based on the GUS interference assay originally developed by the Katagiri lab. It won’t be intuitive to a non specialist that effectors that are recognized suppress GUS expression. Later in the ms the authors discuss using suppression of GUS activity, was this the same assay used in the early part of the paper (eclipse)? If so, the authors should explain it earlier in the paper. I think recognition should be distinguished from the hypersensitive response, which usually means cell death. Its not clear that GUS expression is prevented by cell death versus some other aspect of recognition. The authors mention their surprise at the lack of ion leakage (line 269, 270), but by now there are a number of examples in the literature where recognition occurs without cell death in a resistance gene-dependent manner.

Fig 6

Why is B plotted from Log10 of 0 to 12.5? It makes detecting differences on the graph very difficult. I recommend using no more than log10 of 10 for the y axis. I think that the authors should be cautious about whether the mutations are additive or not, since they did not do a time course of infection. It’s also possible that these effectors are recognized by the same mechanism, which would explain why they are not additive. The authors should do an analysis to compare all strains to each other if they want to say one mutant grows less or more than another. It looks from the graphs that all strains were compared to WT -to decide whether each one was different from WT. Looking at the graph in B for Hort16A, the authors may consider that reduced growth could be due to unmasking of an avirulence effector, which looks most obvious for the strain lacking HopZ5a. Considering that another member of the HopZ family can suppress multiple avirulence effectors, it’s possible that this is what is happening for HopZ5a.

Line 358, suggest deleting “Important”

Reviewer #2: 1) Figure 2C and 2E, please add size references for the DNA ladders, label and indicate the PCR bands with sizes specifically. In Figure legend, it should be (F-H), not (E-H).

2) Line 165-167 and Figure S1, it seems Psa3 X-27 and Psa3 V-13 △sEEL cause severer disease symptom in A. chinensis var. chinensis ‘Hort16A’, compared to Psa3 V-13, which is more pronounced in Psa3 V-13 △sEEL (Figure S1 and Figure S2). Does this suggest that sEEL contributes to the recognition in A. chinensis var. chinensis ‘Hort16A’ as well.

3) Figure S4 legend, please add information for taking samples of how many days post-inoculation.

Reviewer #3: Line 86: Could you clarify non-syntenic in this context? non-syntenic compared to what?

Line 122: Please include a citation for MITEs, mention the DDE repeats and include a citation.

Line 168: Yes ETI could be a factor but there's likely more to host compatibly than failure to elicit ETI.

ETS-failure is the alternative hypothesis.

Line 214: While I appreciate the desire for precision, referencing the two Actinidia species only by their cultivar designations or by the complete species w pathovars designation makes things harder on the reader. Particularly since single cultivars of each are primarily used throughout and cultivar comparisons are not part of this work. I'd suggest either using only "A. arguta' after the first introduction or other practical abbreviations eg Achn Aarg

Line 223: Plasmid loss is one possible explanation but it doesn't seem likely based on the success of the 50-day symptom experiments. This could have been assessed using either plates counts based on the selectable marker or RT-qPCR rather than being speculated. This feels like handwaving rather than a legitimate explanation. It isn't a problem to report observations partially inconsistent with the simple model. Also, partial resistance implies an additive rather than a co-dependence relationship.

Line 333: Is non-host an appropriate term given the circumstances?

Figure 2: The genetic nomenclature used in this manuscript is not intuitive and would be greatly aided by providing a more detailed diagram representing the full Psa3 V13 EEL region the X-27 deletion and the multiple EEL sub-deletions directly in the figure. This does not need to be to scale in the figure but should indicate the gene content. This needs to be in primary figures rather than in the supplemental and should be displayed graphically rather than being keyed in the legend.

Fig S3: diamonds are white and gray

Fig 5: The pastel rainbow color scale system does not work as effectively in the scatter plot since the close colors make it difficult to distinguish. I'd suggest moving Pfo WT results to a supp figure to expand the useful color distinctions.

Fig 7:It would be preferable to indicate presence absence in these specific strains rather than generalized Psa biovar distributions

Reviewer #4: The results of Fig 3B suggest that all four avirulence genes are required for the resistance observed in the Psa3 V-13 wild type strain since individual mutations result in increased growth. Although polymutants do not appear to show cumulative increases in bacterial growth, likely due to compromised virulence, the results of Fig 3B show that the ETI responses are additive since the 4 avirulence genes found in the wild-type strain provides more resistance than the 3 found in the individual mutants. This aspect of additivity should be discussed.

The X-axis of bar graphs should be labeled throughout the manuscript, as done in Figure 3, since it is difficult to follow the colors provided in the Figure Legends (eg. Fig 4, etc…).

Title: Given that strains of this pathovar Psa3 can cause disease on A. arguta the authors should define what they mean by non-host.

The authors should provide background information about the Psa3 strain in the introduction since it is the main focus of the manuscript.

Line 66: What is meant by the “super pan effector repertoire” and how is it different from the “pan effector repertoire”?

Line 139: What is the rationale for the naming the EEL mutants? Does the ‘s’ in sEEL mutant mean anything? Also, the author’s clarify that Psa3 V-13 is the wild-type strain.

Line 161: “Psa3 X-27 and Psa3 V-13 ΔsEEL both produced similar disease symptoms to Psa3 V-13 in ‘Hort16A’…”. It appears that Hort16 is more susceptible to the sEEL mutant (ie. more disease). Does this indicate that the sEEL is also recognized in this variety?

Line 166: “Knocking out the sEEL locus increased virulence in A. arguta AA07_03 quantitatively, but Psa3 X-27 or Psa3 V-13 ΔsEEL were not as virulent in A. arguta AA07_03 as they were in ‘Hort16A’.” This is confusing since Psa3 X-27 appears to be more virulent on AA07_03 than on Hort16A according to FigS2.

Comparing the growth of sEEL on these two varieties supports the authors’ hypothesis that “there may be additional effectors recognized by AA07_03 within the Psa3 V-13 effector complement.”

Line 187: Which strains does “These” refer to?

Line 242: HopF1e reduces bacterial growth when expressed in the sEEL mutant (Fig 4D) but appears to enhance disease symptoms (Fig S8). Why this discrepancy?

PLOS authors have the option to publish the peer review history of their article (what does this mean?). If published, this will include your full peer review and any attached files.

Reviewer #1: No

Reviewer #2: **Yes: **Gang Yu

Reviewer #3: No

Reviewer #4: No
---

## [Decision Letter · Decision Letter 1]

21 Apr 2022

Dear Associate Professor Templeton,

We are pleased to inform you that your manuscript 'Effector loss drives adaptation of Pseudomonas syringae pv. actinidiae biovar 3 to Actinidia arguta' has been provisionally accepted for publication in PLOS Pathogens.

Best regards,

Gitta Coaker, PhD

Guest Editor

PLOS Pathogens

Bart Thomma

Section Editor

PLOS Pathogens

Kasturi Haldar

Editor-in-Chief

PLOS Pathogens

orcid.org/0000-0001-5065-158X

Michael Malim

Editor-in-Chief

PLOS Pathogens

orcid.org/0000-0002-7699-2064

Reviewer Comments (if any, and for reference):

Reviewer's Responses to Questions

**Part I - Summary**

Reviewer #2: Thank the authors addressing the comments.

Reviewer #3: My concerns have been adequately addressed. I support the manuscript in its current form.

Reviewer #4: The authors have addressed my concerns.

**Part II – Major Issues: Key Experiments Required for Acceptance**

Reviewer #2: (No Response)

Reviewer #3: (No Response)

Reviewer #4: (No Response)

**Part III – Minor Issues: Editorial and Data Presentation Modifications**

Reviewer #2: (No Response)

Reviewer #3: (No Response)

Reviewer #4: (No Response)

PLOS authors have the option to publish the peer review history of their article (what does this mean?). If published, this will include your full peer review and any attached files.

Reviewer #2: **Yes: **GANG YU

Reviewer #3: No

Reviewer #4: No

---

## [Editor Report · Acceptance letter]

13 May 2022

Dear Associate Professor Templeton,

We are delighted to inform you that your manuscript, "Effector loss drives adaptation of *Pseudomonas syringae* pv. *actinidiae* biovar 3 to *Actinidia arguta*," has been formally accepted for publication in PLOS Pathogens.

Best regards,

Kasturi Haldar

Editor-in-Chief

PLOS Pathogens

orcid.org/0000-0001-5065-158X

Michael Malim

Editor-in-Chief

PLOS Pathogens

orcid.org/0000-0002-7699-2064